# Trust-Region Diffusion Policies for Massively Parallel On-Policy RL

**Huy Le** [1] [2]   **Onur Celik** [*] [2]   **Denis Blessing** [*] [2]   **Tai Hoang** [*] [2]
**Claas A Voelcker** [3]   **Axel Brunnbauer** [4]   **Felix Richter** [1]   **Michael Volpp** [1]   **Gerhard Neumann** [2]

## Abstract

Reinforcement learning with massively parallel simulations has become a standard framework for developing robust, deployable policies; however, most existing approaches still rely on simple Gaussian policy parameterizations. Diffusion models provide a more expressive policy class and have shown strong performance on challenging control problems, yet most diffusion-based RL methods are designed for offline or off-policy training. In this work, we ask whether diffusion policies can be trained effectively in the massively parallel, on-policy regime. To this end, we introduce Trust-region Diffusion Policies (TruDi), which enables diffusion policies for on-policy RL with massively parallel simulations. This setting is particularly challenging because the data distribution changes quickly across updates, making stable training with complex policies difficult. TruDi addresses this by integrating a trust-region optimization rule to enforce a KL-divergence constraint over the entire diffusion trajectory. Empirically, we evaluate TruDi on a diverse set of 4 massively parallel RL benchmarks comprising a total of 73 tasks. Across these tasks, TruDi consistently outperforms or is on-par with strong baselines on standard tasks and achieves clear gains on more challenging humanoid control tasks, establishing a strong new baseline for massively parallel on-policy RL.

## 1. Introduction

Diffusion models (Ho et al., 2020; Sohl-Dickstein et al., 2015; Song et al., 2021) have shown remarkable results in high-dimensional generative tasks, notably in domains where data from the target distribution is available, e.g., image generation (Ho et al., 2022; Saharia et al., 2022; Rombach et al., 2022) or imitation learning (Chi et al., 2023; Zhou et al., 2024; Carvalho et al., 2025). More recently, their strong representational properties have been explored in reinforcement learning (RL) (Celik et al., 2025; Le et al., 2025; Wang et al., 2024; 2023; Ding et al., 2024; Ren et al., 2024), where the policy is represented by a diffusion model and is trained from scratch. Here, generating an action conditioned on an observation requires first running the diffusion process. The action after the last diffusion time step is then executed in the environment (Wang et al., 2023; Ren et al., 2024; Celik et al., 2025; Le et al., 2025). Diffusion-based policies have been primarily integrated into off-policy RL settings to leverage the framework's data efficiency. This approach has yielded remarkable results with state-of-the-art performance on a wide range of benchmarks.

Recent advances in highly parallelized simulators (Mittal et al., 2025; Zakka et al., 2025; Tao et al., 2025; Hoang et al., 2026) led to a significant acceleration of RL training and to impressive sim-to-real transfer capabilities using Gaussian policy representations (Rudin et al., 2022; Zakka et al., 2025; Kumar et al., 2021; He et al., 2025) in on-policy RL. Despite this breakthrough, training diffusion-based policies from scratch using on-policy RL has not been researched in the literature due to two main reasons. First, training diffusion-based policies is generally more expensive, because several diffusion steps are necessary to generate a single action, and second, essential statistics such as the likelihoods are not easily tractable for diffusion models (Zhou et al., 2024). The latter is particularly important because trust-region constraints have proven essential in on-policy RL methods to avoid premature convergence (Peters et al., 2010a; Schulman et al., 2015; 2017; Abdolmaleki et al., 2015; Hoang et al., 2025), but they require evaluating these essential statistics. Moreover, recent works (Voelcker et al., 2025) have demonstrated that explicitly learning a Q-function in the trust region-constrained maximum entropy objective additionally improves performance in on-policy RL and significantly outperforms PPO (Schulman et al., 2017), which is the common choice among practitioners. Learning this Q-function additionally provides gradient information for updating the policy, which is commonly used

[*]Equal contribution   [1]Bosch Center for Artificial Intelligence [2]Autonomous Learning Robots, KIT [3]University of Texas at Austin [4]TU Wien. Correspondence to: Huy Le <bao-huy.le@de.bosch.com>.

*Proceedings of the $43^{rd}$ International Conference on Machine Learning*, Seoul, South Korea. PMLR 306, 2026. Copyright 2026 by the author(s).

in diffusion-based inference methods (Berner et al., 2022; Vargas et al., 2023; Richter & Berner, 2024; Nusken et al., 2024). These insights motivate the question of whether we can leverage the strong representational capacities of diffusion models in the trust-region constrained maximum entropy on-policy RL setting.

This paper aims to answer this question by proposing Trust-region Diffusion Policies (TruDi). We build upon the probabilistic inference perspective on maximum entropy RL (Toussaint, 2009; Ziebart et al., 2008; Haarnoja et al., 2017; Levine, 2018) and adopt the tractable lower bound for diffusion policies on this objective as proposed by Celik et al. (2025). While Celik et al. (2025) applied this formulation to off-policy learning, in on-policy RL, trust-region constraints have been well established for stable training (Peters et al., 2010b; Schulman et al., 2015; 2017). However, these trust-regions are non-trivial to enforce for diffusion policies due to their intractable marginal likelihoods (Zhou et al., 2024). To address this, we derive a tractable upper bound on the marginal Kullback-Leibler (KL) divergence by constraining the divergence between the entire diffusion trajectories of the current and behavior policies. Constraining this upper bound effectively enforces a trust region over the whole generation process, ensuring the stability required for massively parallel on-policy learning. The resulting algorithm is a trust-region constrained on-policy RL method that is sample-efficient and requires only marginally longer wall-clock training time compared to Gaussian counterparts. Additionally, we demonstrate how to leverage the probability flow ODE (Song et al., 2021) to generate high-return actions with a higher likelihood compared to those obtained using the SDE and other evaluation techniques in diffusion-based policies.

To summarize, our contributions are threefold: **(i)** we introduce a principled framework for training diffusion policies in the on-policy, massively parallel RL setting by enforcing a trust-region constraint over the full diffusion trajectory; **(ii)** we provide a comprehensive empirical evaluation across standard continuous-control and large-scale robotic benchmarks, showing that our method is competitive with strong Gaussian on-policy baselines on standard tasks while delivering clear gains on challenging high-dimensional humanoid control; and **(iii)** we present detailed analyses of key design choices, including the effect of the trust-region threshold, different sampling strategies for the policy evaluation (e.g., SDE/ODE/best-of-$K$), and multimodality on symmetric tasks, highlighting which components are most important for stable and effective training.

## 2. Related Work

**Massively Parallel RL.** The advent of GPU-accelerated simulators (Mittal et al., 2025; Tao et al., 2025; Hoang et al.,

2026) has shifted the computational bottleneck from data generation to policy learning. To leverage this throughput, research initially focused on scaling off-policy algorithms: pioneering efforts like Parallel Q-Learning (Li et al., 2023) decoupled actor and learner processes, while recent state-of-the-art methods such as FastTD3 (Seo et al., 2025b) and FastSAC (Seo et al., 2025a) utilize large-batch updates with on-GPU replay buffers to minimize training latency. However, despite their speed, these off-policy approaches incur significant memory overheads, as maintaining large buffers in memory constrains the capacity for parallel environments and large network architectures. This has renewed interest in on-policy methods, which minimize memory footprint by consuming data immediately. This lineage spans from constrained optimization formulations like REPS (Peters et al., 2010a) and MORE (Abdolmaleki et al., 2015) to scalable deep RL approximations like PPO (Schulman et al., 2017) and differentiable projection layers (Otto et al., 2021; Li et al., 2024a). Building on these foundations, REPPO (Voelcker et al., 2025) demonstrated that rigorously enforcing trust-region constraints via dual ascent allows on-policy RL to match off-policy sample efficiency without the associated memory bottlenecks. While this primal-dual framework is straightforward for Gaussian policies with exact action likelihoods, applying it to diffusion policies is fundamentally bottlenecked by their intractable marginals. TruDi overcomes this barrier by introducing a novel trajectory-level trust region, successfully enabling highly expressive, multimodal diffusion policies for massively parallel on-policy RL. Synthesizing these directions, REPPO (Voelcker et al., 2025) stabilizes pathwise policy gradients via primal-dual trust-region updates, allowing on-policy RL to match off-policy sample efficiency without the memory bottlenecks. While this framework is restricted to Gaussian policies due to the need for tractable action likelihoods, TruDi overcomes this bottleneck by introducing a novel trajectory-level trust region, enabling expressive diffusion policies for massively parallel on-policy RL.

**Diffusion-based policies in RL.** Early research on diffusion models in reinforcement learning primarily focused on the offline setting (Wang et al., 2023; Janner et al., 2022; Chen et al., 2023). These works utilized diffusion models either as high-fidelity trajectory generators (Janner et al., 2022) or as expressive policy priors to regularize behavior in static datasets (Hansen-Estruch et al., 2023; Kang et al., 2023; Lu et al., 2023; Mao et al., 2024; Fang et al., 2025). The success of these methods catalyzed the development of online diffusion-based RL. Initial approaches, such as DIPO (Yang et al., 2023) and its multimodal extension (Li et al., 2024b), employed behavior cloning updates with Q-function guidance but relied on intrinsic stochasticity for exploration. Subsequent methods like QSM (Psenka et al., 2024) and QVPO (Ding et al., 2024) streamlined optimiza-

tion by matching scores or weighting diffusion losses with Q-values. However, these methods often disregarded policy entropy, necessitating ad-hoc exploration heuristics like Gaussian noise injection (Psenka et al., 2024) or uniform sampling (Ding et al., 2024). DACER (Wang et al., 2024) attempted to address this with an entropy regulator but relied on approximate Gaussian Mixture Models. DIME (Celik et al., 2025) introduces diffusion models into the Maximum Entropy RL framework for continuous control. In parallel, HyDo (Le et al., 2025) leveraged maximum entropy formulation for hybrid action spaces in manipulation tasks. However, despite their theoretical rigor, these methods generally operate in the *off-policy* regime. Their reliance on large experience replay buffers creates substantial memory bottlenecks, making it difficult to scale to the massively parallel simulation environments required for modern robot learning (Seo et al., 2025b;a).

To overcome these scalability bottlenecks, a recent wave of works has shifted toward the *on-policy* setting. One line of research retains the diffusion formulation, adapting trust-region or mirror descent methods to handle stochastic chains. For instance, DPPO (Ren et al., 2024) fine-tunes diffusion policies using PPO, while GenPO (Ding et al., 2025) approximates likelihoods to enable learning from scratch. This direction has also extended to discrete combinatorial spaces, where Ma et al. (2025) utilized Policy Mirror Descent to optimize discrete diffusion policies. A parallel direction leverages flow matching to bypass the intractability of diffusion likelihoods, as seen in FPO (McAllister et al., 2025), FlowRL (Lv et al., 2025), and ReinFlow (Zhang et al., 2026). However, standard flow-based methods often lack explicit entropy regularization, and concurrent diffusion approaches like DPPO (Ren et al., 2024) and GenPO (Ding et al., 2025) rely on heuristic per-step clipping or complex invertible mappings. TruDi derives a principled trust-region formulation over the full diffusion trajectory within the Maximum Entropy RL framework, utilizing a tractable trajectory-level KL bound to enable stable, Q-based pathwise policy updates.

## 3. Preliminaries

**Notation.** We consider a Markov decision process defined by the tuple $(\mathcal{S}, \mathcal{A}, r, p, \rho_\pi, \gamma)$, where we aim to optimize a probabilistic policy $\pi_\theta : \mathcal{S} \times \mathcal{A} \to \mathbb{R}^+$ that is defined in continuous state and action spaces denoted by $\mathcal{S}$ and $\mathcal{A}$, respectively, and is parameterized with parameters $\theta$. The objective function $J(\pi_\theta)$ is defined as the discounted sum of expected future rewards, where $\gamma \in [0, 1)$ denotes the discount factor, and $r : \mathcal{S} \times \mathcal{A} \to [r_{\min}, r_{\max}]$ is a bounded reward function. In state $s_t \in \mathcal{S}$ at time step $t$, the agent transitions to state $s_{t+1} \in \mathcal{S}$ by executing an action $a_t$ that is sampled from the policy $\pi_\theta$. The probability

density of transitioning to the state $s_{t+1}$ is denoted by $p : \mathcal{S} \times \mathcal{S} \times \mathcal{A} \to \mathbb{R}^+$. We follow prior work (Haarnoja et al., 2018a) and overload the notation for the state and state-action distributions induced by the policy by referring to both distributions as $\rho_\pi$.

**Maximum Entropy Reinforcement Learning.** To control the exploration-exploitation tradeoff, maximum entropy RL (MaxEnt RL) (Ziebart et al., 2008; Toussaint, 2009; Haarnoja et al., 2017) augments the reward in each time step with the entropy of the current policy $\mathcal{H}(\pi_\theta(a \mid s)) = -\int \pi_\theta(a \mid s) \log \pi_\theta(a \mid s) \, \mathrm{d}a$, resulting in the objective

$$J(\pi_\theta) = \sum_{t=0}^{\infty} \gamma^t \mathbb{E}_{\rho_\pi} \left[ r_t + \alpha \mathcal{H}(\pi_\theta(a_t \mid s_t)) \right], \quad (1)$$

where $\alpha \in [0, \infty)$ is a entropy scaling factor. Note that we denote the reward at time step $t$ as $r_t \triangleq r(s_t, a_t)$. We similarly define the entropy-augmented Q-function under the current policy $\pi_\theta$ as

$$Q^{\pi_\theta}(s_t, a_t) = r_t + \sum_{l=1}^{\infty} \gamma^l \mathbb{E}_{\rho_\pi} \left[ r_{t+l} \right.$$
$$\left. + \alpha \mathcal{H}(\pi_\theta(a_{t+l} \mid s_{t+l})) \right]. \quad (2)$$

In the context of policy iteration, the policy improvement step can be formulated as an approximate inference problem (Haarnoja et al., 2017). Given the Q-function of the previous policy $\pi_{\mathrm{old}}$, we seek to find a new policy $\pi_\theta$ that minimizes the KL divergence to the Boltzmann distribution induced by $Q^{\pi_{\mathrm{old}}}$:

$$\mathcal{L}(\pi_\theta) = D_{\mathrm{KL}} \left( \pi_\theta(a_t \mid s_t) \, || \, \frac{\exp(Q^{\pi_{\mathrm{old}}}(s_t, a_t)/\alpha)}{\mathcal{Z}^{\pi_{\mathrm{old}}}(s_t)} \right), \quad (3)$$

where $\mathcal{Z}^{\pi_{\mathrm{old}}}(s_t) = \int \exp(Q^{\pi_{\mathrm{old}}}(s_t, a_t)/\alpha) \, \mathrm{d}a$ is the partition function. Minimizing this KL divergence projects the parametric policy $\pi_\theta$ onto the target Boltzmann distribution, thereby guaranteeing an improvement in the MaxEnt objective in Eq. 1.

**Denoising Diffusion Policy.** We consider the state-conditioned variance preserving (VP) stochastic differential equation (SDE) in which the *forward* or *noising process* is given by an Ornstein-Uhlenbeck (OU) process (Särkkä & Solin, 2019) defined by the SDE

$$\mathrm{d}a^\tau = -\beta^\tau a^\tau \mathrm{d}\tau + \eta\sqrt{2\beta^\tau}\mathrm{d}B^\tau, \quad a^0 \sim \vec{\pi}^0(\cdot \mid s), \quad (4)$$

with Brownian motion $(B^\tau)^{\tau \in [0,T]}$, diffusion coefficient $\beta : [0, T] \to \mathbb{R}^+$ and the prior distribution's standard deviation $\eta$.[1] Given a state $s$, the forward process starts from a target

---

[1]Note that we deviate from the standard notation and denote the diffusion time parameter by superscript $\tau$ here, in order to distinguish from the MDP time step which we denoted by the subscript $t$.

policy $a^0 \sim \vec{\pi}^0(\cdot \mid s)$ at $\tau = 0$ and continuously adds noise, such that (for large enough $T$) the marginal distribution defined by the SDE is given by $\vec{\pi}^T(\cdot \mid s) \approx \mathcal{N}(0, \eta^2 I)$.

The *backward* (or *generative*) *process* corresponding to the SDE in Eq. 4 is given by

$$\mathrm{d}a^\tau = \left(-\beta^\tau a^\tau - 2\eta^2 \beta^\tau \nabla_a \log \vec{\pi}^\tau(a^\tau \mid s)\right) \mathrm{d}\tau \\ + \eta\sqrt{2\beta^\tau} \mathrm{d}B^\tau, \quad (5)$$

which starts from Gaussian noise at $\tau = T$ and gradually transforms it into samples from the target action distribution at $\tau = 0$. In practice, the score $\nabla_a \log p^\tau(a^\tau \mid s)$ is unknown and approximated by a neural network $f_\theta^\tau(a^\tau, s)$. After training, simulating the reverse process yields a sample $a^0$, which we execute as the action $a \sim \pi_\theta(\cdot \mid s)$.

**Discretizing the Diffusion Policy.** The discretization implies Gaussian transition kernels between successive diffusion steps, and the corresponding trajectory distributions factorize as

$$\vec{\pi}^{0:N}(a^{0:N} \mid s) = \vec{\pi}^0(a^0 \mid s) \prod_{n=0}^{N-1} \vec{\pi}^{n+1|n}(a^{n+1} \mid a^n, s), \quad (6)$$

$$\overleftarrow{\pi}_\theta^{0:N}(a^{0:N} \mid s) = \overleftarrow{\pi}^N(a^N \mid s) \prod_{n=1}^{N} \overleftarrow{\pi}_\theta^{n-1|n}(a^{n-1} \mid a^n, s), \quad (7)$$

where $\overleftarrow{\pi}^N(\cdot \mid s)$ denotes the (fixed) Gaussian prior at diffusion step $N$. The executed action distribution is the marginal of the reverse chain, $\pi_\theta(a \mid s) \equiv \overleftarrow{\pi}_\theta^0(a^0 \mid s)$, obtained by integrating out the latent trajectory $a^{1:N}$. These tractable trajectory factorizations will be central in the next section, where we derive tractable bounds for MaxEnt policy improvement and trust-region constraints in joint trajectory space.

## 4. Trust-region Diffusion Policies

A common approach for preventing premature convergence and stabilizing the training in on-policy reinforcement learning is employing a trust region on the policy $\pi_\theta$ (Peters et al., 2010b; Schulman et al., 2015; Otto et al., 2021; Voelcker et al., 2025), leading to the constrained optimization problem

$$\max_\theta \quad \sum_{t=0}^{\infty} \gamma^t \mathbb{E}_{\rho_\pi} \left[ r_t + \alpha \mathcal{H}(\overleftarrow{\pi}_\theta^0(a_t^0 \mid s_t)) \right] \quad (8)$$

$$\text{s.t.} \quad \mathbb{E}_{\rho^\pi} \left[ D_{\mathrm{KL}} \left( \overleftarrow{\pi}_{\mathrm{old}}^0(a_t^0 \mid s_t) \mid\mid \overleftarrow{\pi}_\theta^0(a_t^0 \mid s_t) \right) \right] \leq \epsilon, \quad (9)$$

where the policy parameters $\theta$ are updated such that the maximum entropy RL objective is maximized under the

constraint that the KL divergence between the old policy $\overleftarrow{\pi}_{\mathrm{old}}^0 = \overleftarrow{\pi}_{\theta_{\mathrm{old}}}^0$ and the current policy $\overleftarrow{\pi}_\theta^0$ is bounded by $\epsilon \in \mathbb{R}^+$. Both the objective in Eq. 8 and the constraint in Eq. 9 require calculating the likelihood of the marginal distribution $\pi_\theta^0(a_t \mid s_t)$, which is hard to calculate for diffusion models (Zhou et al., 2024).

To guide the reader, we briefly outline the main steps of this section. We first interpret the denoising process (Eq. 7) as a latent-variable model (Luo, 2022; Ho et al., 2020), which yields a tractable lower bound on the objective. Based on this view, we derive a tractable upper bound for the KL constraint in Eq. 9 and use it to formulate our final optimization problem. We then describe the resulting policy update and parameter learning procedure for TruDi. Finally, we introduce a deterministic evaluation scheme based on the probability-flow ordinary differential equation (ODE) associated with the diffusion policy.

### 4.1. Diffusion Policies as Latent Variable Models in MaxEnt RL

The discretized processes in Eq. 6 and Eq. 7 motivate viewing diffusion policies as latent variable models (Luo, 2022) in which the final action $a^0$ of the diffusion process is the result of sampling from the marginal policy

$$\overleftarrow{\pi}_\theta^0(a^0 \mid s) = \int \overleftarrow{\pi}_\theta^{0:N}(a^{0:N} \mid s) \mathrm{d}a^{1:N}, \quad (10)$$

where $a^{1:N}$ are considered latent variables. Evaluating the likelihood in Eq. 10 is not straightforward, which renders the commonly used approximate inference scheme (Eq. 3) for maximum entropy RL intractable for diffusion-based policies. However, we can obtain a tractable upper bound by applying the data processing inequality (Cover, 1999)

$$D_{\mathrm{KL}} \left( \overleftarrow{\pi}_\theta^0(a^0 \mid s) \mid\mid \vec{\pi}^0(a^0 \mid s) \right) \\ \leq D_{\mathrm{KL}} \left( \overleftarrow{\pi}_\theta(a^{0:N} \mid s) \mid\mid \vec{\pi}(a^{0:N} \mid s) \right), \quad (11)$$

where $\vec{\pi}^0(a^0 \mid s) = \frac{\exp Q_\phi^{\overleftarrow{\pi}}(s, a^0)/\alpha}{\mathcal{Z}^{\overleftarrow{\pi}}(s)}$. Instead of matching the action marginals (LHS of the inequality), this upper bound provides an objective to match the denoising (Eq. 6) with the noising process (Eq. 7). In other words, aligning the denoising process $\overleftarrow{\pi}_\theta(a^{0:N}|s)$ with the noising process $\vec{\pi}(a^{0:N}|s)$ minimizes the approximate inference objective and hence allows maximizing the maximum entropy RL objective in Eq. 8. (Celik et al., 2025; Berner et al., 2022; Nusken et al., 2024) demonstrated that the upper bound in Eq. 11 leads to the lower bound on the marginal entropy $\mathcal{H}(\overleftarrow{\pi}_\theta^0(a^0|s)) \geq l_{\overleftarrow{\pi}_\theta}(a^0, s)$, where

$$l_{\overleftarrow{\pi}_\theta}(a^0, s) = \mathbb{E}_{\overleftarrow{\pi}_\theta^{0:N}} \left[ \log \frac{\vec{\pi}^{1:N|0}(a^{1:N} \mid a^0, s)}{\overleftarrow{\pi}_\theta^{0:N}(a^{0:N} \mid s)} \right], \quad (12)$$

which redefines the Q function for diffusion-based policies

$$Q_\phi^{\bar{\pi}}(s_t, a_t^0) = r_t + \sum_{l=1} \gamma^l \mathbb{E}_{\rho_\pi} \left[ r_{t+l} + \alpha l_{\bar{\pi}_\theta}(a_{t+l}^0, s_{t+l}) \right].$$

(13)

Crucially, this latent variable model view allows us to impose trust-region constraints directly on the joint trajectory distributions, yielding a tractable and principled alternative to marginal KL constraints (Eq. 9), which are otherwise intractable for diffusion policies. We develop this idea in the next section.

### 4.2. Enforcing Trust-Region for Diffusion Policies

Similar to the entropy in Eq. 8, the trust region in Eq. 9 requires evaluating the marginal likelihood (Eq. 10), which is not straightforward to calculate. However, instead of constraining the policy update using Eq. 9, which measures the KL between the marginal action distributions, we propose using an upper bound that measures the information loss on the joint distributions of the denoising processes

$$D_{\text{KL}} \left( \bar{\pi}_{\text{old}}^0(a^0 \mid s) \| \bar{\pi}_\theta^0(a^0 \mid s) \right)$$

(14)

$$\leq D_{\text{KL}} \left( \bar{\pi}_{\text{old}}^{0:N}(a^{0:N} \mid s) \mid \bar{\pi}_\theta^{0:N}(a^{0:N} \mid s) \right).$$

A derivation is provided in Appendix A. This upper bound can now easily be approximated by samples as the likelihoods are w.r.t. to the tractable joint distribution (Eq. 7).

Using the upper bounds for the objective (Eq. 11) and the constraint (Eq. 14), we can now formulate our final optimization problem as

$$\min_\theta \quad \mathbb{E}_{\rho^\pi} \left[ D_{\text{KL}} \left( \bar{\pi}_\theta^{0:N}(a^{0:N} \mid s) \| \bar{\pi}^{0:N}(a^{0:N} \mid s) \right) \right]$$

(15)

$$\text{s.t.} \quad \mathbb{E}_{\rho^\pi} \left[ D_{\text{KL}} \left( \bar{\pi}_{\text{old}}^{0:N}(a^{0:N} \mid s) \| \bar{\pi}_\theta^{0:N}(a^{0:N} \mid s) \right) \right] \leq \epsilon.$$

(16)

### 4.3. Policy and Critic Optimization

We are now ready to describe TruDi update rules. Here, building on the formulation in Section 4.2, we follow the standard actor-critic policy iteration framework from (Haarnoja et al., 2018a) and adapt it to the on-policy setting, as in REPPO (Voelcker et al., 2025).

**Fitting the Q-function.** For optimizing the parameters $\phi$ of the Q-function in Eq. 13, we rely on recent insights for on-policy RL methods (Voelcker et al., 2025). More precisely, we employ TD-$\lambda$ (Sutton, 1988) for generating the soft target values corresponding to the current policy $\pi_\theta$:

$$G^\lambda(s, a) = \frac{1}{\sum_{n=0}^N \lambda^n} \sum_{n=0}^N \lambda^n G^{(n)}(x, a). \quad (17)$$

This represents a Monte-Carlo estimate of the soft Q-function generated with the data from the current policy $\pi_\theta$. Here, $G^{(n)}$ is defined as

$$G^{(n)}(s_t, a_t) = \sum_{k=t}^{n-1} \gamma^k \left( r(s_k, a_k) - \alpha \log \pi^N(a^N \mid s_k) \right.$$

$$\left. - \alpha \sum_{n=1}^N \log \frac{\bar{\pi}^{n|n-1}(a_k^n \mid a_k^{n-1}, s_k)}{\bar{\pi}_\theta^{n-1|n}(a^{n-1} \mid a^n, s_k)} \right) + \gamma^n Q(s_n, a_n).$$

(18)

For fitting the Q-function's parameters $\phi$, we use HL-Gauss (Imani & White, 2018; Farebrother et al., 2024), which relies on the cross-entropy loss function rather than the known squared Bellman residual that is prone to outliers (Farebrother et al., 2024; Voelcker et al., 2025).

**Policy Update.** For a fixed Lagrangian multiplier $\lambda$ (distinct from the TD-$\lambda$ trace parameter used above) to the constraint in Eq. 16 and a fixed entropy-scaling parameter $\alpha$, recent on-policy approaches (Voelcker et al., 2025) have suggested a split objective

$$\mathcal{L}_{\text{TR}}(\theta \mid \lambda, \alpha, \{s_i\}_{i=1}^B) = \frac{1}{B} \sum_{i=1}^B \begin{cases} h(s_i, a), & \text{if } c(s_i) \leq \epsilon \\ g(s_i, \lambda), & \text{otherwise.} \end{cases}$$

(19)

This objective first checks whether the trust region is violated by evaluating

$$c(s_i) = \frac{1}{K} \sum_{j=1}^K \sum_{n=1}^N \log \frac{\bar{\pi}_{\text{old}}^{n-1|n}(a_j^{n-1} \mid a_j^n, s_i)}{\bar{\pi}_\theta^{n-1|n}(a_j^{n-1} \mid a_j^n, s_i)},$$

which is a sample-based approximation of the trust region upper bound Eq. 14 with $K$ action samples $a \sim \pi_\theta(\cdot \mid s_i)$ from the current policy. For the case $c(s_i) \leq \epsilon$ (trust region not violated), this objective considers the lower-bound to the maximum entropy RL objective

$$h(s_i, a^0) = \alpha \log \bar{\pi}^N(a^N \mid s_i) - Q_\phi^{\bar{\pi}}(s, a^0)$$

$$+ \alpha \sum_{n=1}^N \log \frac{\bar{\pi}_\theta^{n-1|n}(a^{n-1} \mid a^n, s)}{\bar{\pi}^{n|n-1}(a^n \mid a^{n-1}, s)},$$

(20)

Recall that $a^0 \sim \pi_\theta(\cdot | s_i)$. Finally, for the case $c(s_i) > \epsilon$ (trust region violated), the parameters $\theta$ of the policy are updated purely based on

$$g(s_i, \lambda) = \lambda c(s_i),$$

where $c(s_i)$ is the trust-region estimate from Eq. 20.

**Dual Parameter Updates.** Following recent actor-critic RL frameworks (Haarnoja et al., 2018b; Voelcker et al., 2025), we auto-tune the entropy temperature $\alpha$ toward a target entropy $\epsilon_{\bar{H}}$, and we update the Lagrangian multiplier

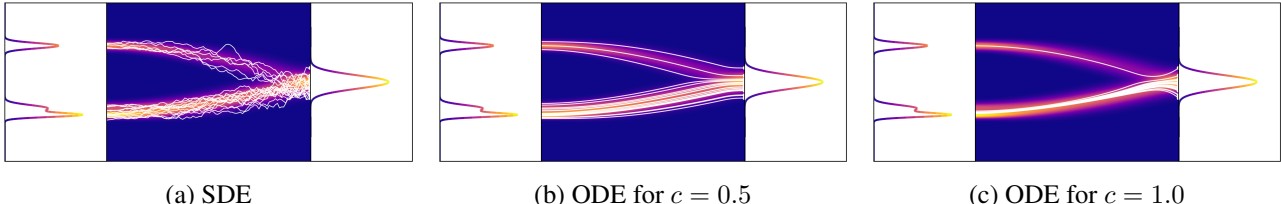

| (a) SDE | (b) ODE for $c = 0.5$ | (c) ODE for $c = 1.0$ |

*Figure 1.* **Scaling the score function modifies the marginal distributions in each time step, inducing greedier sampling at higher scaling values.** Fig. **(a)** visualizes the trajectories (white) of the (unscaled) denoising process starting at the Gaussian prior (right) and generating samples from the target distribution (left). **(b)** visualizes the trajectories (white) of the ODE whose marginal distributions are the same as the SDE in (a). This alignment results from scaling the score with $c = 0.5$ (Song et al., 2021). Higher values for $c$ result in sharper marginal distributions, leading to more greedy samples, as shown by the trajectories of the ODE for $c = 1.0$ in **(c)**. During the evaluation of a diffusion-based policy, simulating the ODE with a scaled score function leads to higher returns.

$\lambda$ associated with the trust-region constraint in Eq. 16 using the dual updates

$$\alpha \leftarrow \alpha - \eta_\alpha \nabla_\alpha \mathbb{E}_{\rho^\pi} \left[ l_{\bar{\pi}_\theta}(a^0, s) - \epsilon_{\bar{H}} \right] \quad (21)$$

$$\lambda \leftarrow \lambda - \eta_\lambda \nabla_\lambda \mathbb{E}_{\rho^\pi} \left[ D_{\text{KL}}\big(\overleftarrow{\pi}_{\text{old}}^{0:N} \,\|\, \overleftarrow{\pi}_\theta^{0:N}\big) - \epsilon \right], \quad (22)$$

where we denote $\overleftarrow{\pi}_\theta^{0:N} = \overleftarrow{\pi}_\theta^{0:N}(a^{0:N}|s)$ and $\overleftarrow{\pi}_{\text{old}}^{0:N} = \overleftarrow{\pi}_{\text{old}}^{0:N}(a^{0:N}|s)$ in the dual update, respectively. Although iteratively updating $\phi, \theta$ and $\alpha, \lambda$ does not guarantee that the constraints are satisfied, this dual descent strategy has empirically shown a sufficiently fast adaptation of the Lagrangian multipliers while being easy to implement in practice.

### 4.4. Probability-Flow ODE for Policy Evaluation

During evaluation, we aim to generate actions deterministically that are likely to achieve high returns. For Gaussian policies, a natural deterministic representative is the mean action; for diffusion policies, actions are obtained through an iterative denoising process, so an analogous procedure is less obvious. We therefore use the probability-flow ODE associated with the reverse diffusion process (Song et al., 2021). Using an Euler discretization, we obtain

$$a^{n-1} = a^n + \big(\beta^n a^n + c2\,\eta^2\beta^n \nabla \log \overrightarrow{\pi}^n(a^n \mid s)\big)\,\delta, \quad (23)$$

where $c \in \mathbb{R}^+$ scales the score term. For $c = \frac{1}{2}$, the probability-flow dynamics form the deterministic counterpart to the following SDE (Särkkä & Solin, 2019)

$$a^{n-1} = a^n + \big(\beta^n a^n + 2\eta^2\beta^n \nabla \log \overrightarrow{\pi}^n(a^n \mid s)\big)\,\delta + \xi^n, \quad (24)$$

and match its marginals (Song et al., 2021). This alignment is also reflected by the similar behavior of the stochastic SDE trajectories in Figure 1a and the deterministic ODE trajectories in Figure 1b.

While matching marginals is desirable, it does not necessarily concentrate the deterministic trajectory on the highest-

return regions. In practice, we use $c > \frac{1}{2}$ as an evaluation-only heuristic that biases trajectories toward higher-density regions and empirically increases the likelihood of high-return actions ( Figure 1c). Intuitively, scaling the score can be interpreted as tempering intermediate marginals since

$$\nabla \log \big(\overrightarrow{\pi}^n(a^n \mid s)\big)^c = c\,\nabla \log \overrightarrow{\pi}^n(a^n \mid s), \quad (25)$$

so larger $c$ sharpens the distribution and smaller $c$ smooths it. Note that $c \neq \frac{1}{2}$ changes the score field (Song et al., 2021). However, we apply this modification only at evaluation time and do not update the score network.

## 5. Experimental Setup

We evaluate TruDi against state-of-the-art on-policy baselines, covering both Gaussian policies (PPO (Schulman et al., 2017), REPPO (Voelcker et al., 2025), SPO (Xie et al., 2025)) and diffusion-based (or flow-based) policies (DIME (Celik et al., 2025), DPPO (Ren et al., 2024), FPO (McAllister et al., 2025)). See Section B.5 for details. We conduct experiments on three commonly used RL benchmark suites: 20 tasks from the MuJoCo Playground DMC suite, 13 ManiSkill environments (Tao et al., 2025), 30 Humanoid-Bench environments (Sferrazza et al., 2024), and 6 environments in IsaacLab (Mittal et al., 2025). ManiSkill, MuJoCo Playground and IsaacLab provide GPU-accelerated simulation, which is particularly well suited for on-policy RL methods. These benchmarks include a wide range of tasks, from classical control to robotic manipulation and locomotion. To further stress-test the benefit of diffusion policies, we additionally evaluate TruDi on Humanoid-Bench (Sferrazza et al., 2024), a recent benchmark consisting of 36 challenging whole-body manipulation tasks on a humanoid.

We report average returns and, when available, success rates using the interquartile mean (IQM) with 95% confidence intervals over 10 seeds for all methods as suggested by (Agarwal et al., 2021).

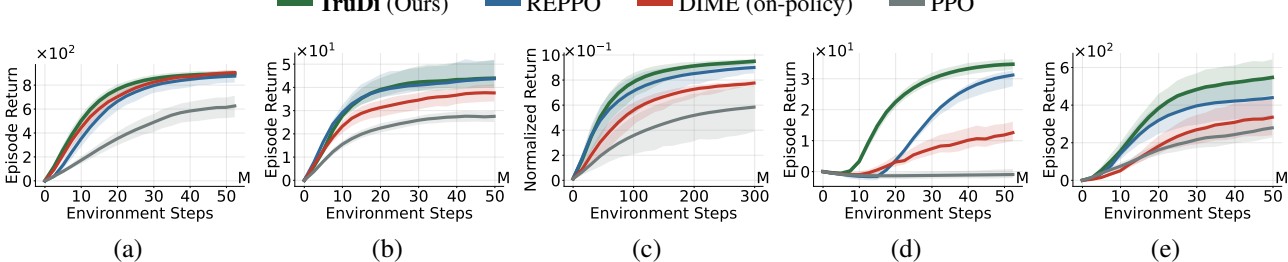

*Figure 2.* **Aggregated Performance on Continuous Control Benchmarks.** We compare TruDi against strong on-policy baselines (REPPO, PPO) and the diffusion-based online RL method DIME. Curves depict the *Interquartile Mean (IQM)* of episode returns with 95% stratified bootstrap confidence intervals. The evaluation spans five distinct benchmarks: **(a)** Standard continuous control (**MuJoCo Playground DMC**, 20 tasks); **(b)** Fine-grained manipulation (**ManiSkill3**, 14 tasks); **(c)** Robust locomotion & dexterity (**IsaacLab**, 6 tasks, normalized returns); **(d)** Humanoid locomotion (**MuJoCo Playground**, 4 tasks); and **(e)** Whole-body control (**HumanoidBench**, 27 tasks). TruDi matches or exceeds the performance of strong on-policy baselines (REPPO) on standard control tasks (a-c) while significantly outperforming baselines on difficult, high-dimensional humanoid tasks (d-e).

# 6. Results

**Standard Benchmarks.** On the standard RL benchmarks including the MuJoCo Playground DMC suite (Figure 2a), ManiSkill (Figure 2b), and IsaacLab (Figure 2c), TruDi consistently outperforms all baselines throughout training and achieves the best final performance. In these settings, TruDi and REPPO form a clear top tier, while the remaining methods often converge early and appear to get stuck in sub-optimal regions. Compared to REPPO, TruDi converges to a higher asymptotic return on both DMC and IsaacLab with a visible gap in the final performance. DIME also performs strongly on the DMC suite, consistent with the results reported in the original paper, but its performance drops when scaling to the high-dimensional, contact-rich environments present in our experiments.

**Complex Humanoid Tasks.** We next move to the humanoid benchmarks, namely the MuJoCo Playground Humanoid tasks and Humanoid-Bench, which require high-dimensional whole-body control with long-horizon behaviors and complex coordination. Figure 2 (d,e) show a clear gap between TruDi and its main competitor REPPO, with TruDi achieving substantially higher episode return on both suites. This highlights the benefit of combining diffusion policies with our trust-region update, which improves exploration while remaining stable in these difficult settings, whereas the adaptive on-policy diffusion approach from DIME often fails to make consistent progress in our massively parallel setup. Looking at the per-task results in Appendix Figure 13 and 14, the advantage becomes most visible on the hardest environments such as window cleaning, balancing, hurdling, and cube, where Gaussian baselines frequently plateau early, and TruDi continues to improve and reaches higher final performance. Crucially, these results also support the stability of our method, since it matches tuned Gaussian policies on standard domains, but uses the

added expressivity of diffusion policies when the action distribution becomes more complex in humanoid control.

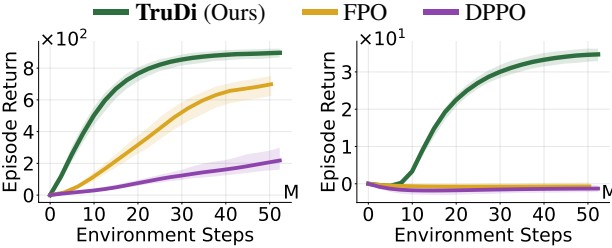

*Figure 3.* Evaluation on Diffusion/Flow based Policies. **Left:** Mujoco Playground DMC. **Right:** Mujoco Playground Humanoid.

**Comparison with Diffusion and Flow-Based Baselines.** We evaluate TruDi against two state-of-the-art methods that also utilize expressive generative policies: DPPO (Ren et al., 2024), which applies PPO-style clipping to diffusion policies, and FPO (McAllister et al., 2025), which utilizes Flow Matching for policy parameterization. Figure 3 presents the aggregated learning curves on the MuJoCo Playground DMC (Left) and Humanoid (Right) benchmarks. On *DMC suite*, TruDi achieves comparable or superior sample efficiency to FPO and significantly outperforms DPPO. This performance gap becomes more pronounced on the high-dimensional *Humanoid tasks* (Right).

**Multimodality of Diffusion Policies.** To empirically validate the capacity of TruDi to represent multimodal action distributions, we evaluate it on the PushT and StackCube tasks. As shown in Figure 4, these environments are designed with symmetric optimal solutions (e.g., stacking Red-on-Blue vs. Blue-on-Red) inspired by the object manipulation setups in ManiSkill3 (Tao et al., 2025), creating a bimodal optimization problem. To quantify this multimodality, we adopt the Behavior Entropy metric (Jia et al., 2024).

*Table 1.* Multimodality Analysis. Evaluation of solution diversity on symmetric tasks (200 deterministic runs). Behavior Entropy ($^*$) (see definition in Appendix Section D.3) is normalized with 1 indicating a uniform distribution and 0 indicating mode collapse. As expected, diffusion policies such as TruDi and DIME can capture multiple modes while the pure-Gaussian policy collapses. Best results are highlighted with orange.

| Task | Method | Entropy$^*$ | Episode Return |
|------|--------|-------------|----------------|
| PushT | REPPO | 0 | 154.92 ± 5.85 |
| | DIME | 0.20 ± 0.20 | 171.92 ± 1.08 |
| | TruDi | 0.32 ± 0.13 | 167.32 ± 4.42 |
| StackCube | REPPO | 0 | 83.92 ± 0.24 |
| | DIME | 0.68 ± 0.15 | 84.40 ± 0.69 |
| | TruDi | 0.87 ± 0.08 | 84.69 ± 0.10 |

Let $\mathcal{B}$ represent the discrete set of successful behavior modes ($|\mathcal{B}| = 2$ for our symmetric tasks). The normalized entropy is computed over the empirical behavior distribution $\pi(\beta)$ as:

$$\mathcal{H}(\pi(\beta)) = -\sum_{\beta \in \mathcal{B}} \pi(\beta) \log_{|\mathcal{B}|} \pi(\beta)$$

This scales the metric strictly into the $[0, 1]$ interval, where 0 indicates complete mode collapse and 1 indicates perfectly balanced diversity. We conduct 200 evaluation runs from a fixed initial state, using deterministic generation (ODE for diffusion, mean for Gaussian) to ensure that observed diversity arises from learned modes rather than stochastic noise. As shown in Table 1, the Gaussian-based baseline, REPPO, exhibits zero entropy across both tasks, indicating a complete collapse to a single deterministic solution. In contrast, TruDi achieves high entropy (0.32 and 0.87), confirming that it successfully leverages the expressivity of diffusion models to capture multiple valid solutions in the on-policy setting without sacrificing performance.

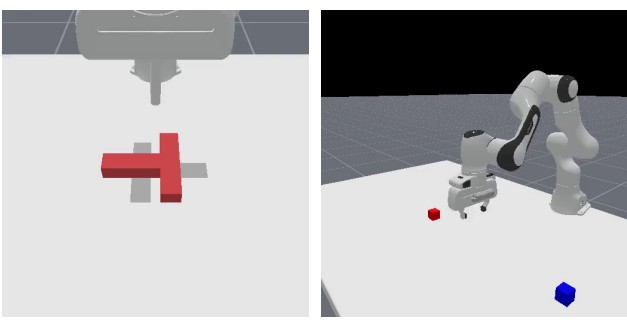

*Figure 4.* Environments for evaluating multimodal behavior. **Left**: PushT, where rotating a T-shaped block 180 degrees to a goal orientation is optimally solved either clockwise or counter-clockwise. **Right**: StackCube, where stacking red-on-blue or blue-on-red are both optimal strategies. Both configurations create bimodal equivalent solutions.

*Table 2.* Computational Efficiency. Wall-clock time and aggregated performance after 50M steps on MuJoCo Playground Humanoid tasks. TruDi achieves the highest performance with a manageable increase in training duration. Best results are highlighted with orange.

| Method | Training Time (Hours) | Episode Return (IQM) |
|--------|----------------------|----------------------|
| PPO | 0.95 ± 0.35 | 0.1 ± 3.7 |
| REPPO | 1.07 ± 0.35 | 29.6 ± 7.2 |
| DIME | 1.38 ± 0.36 | 15.6 ± 9.4 |
| **TruDi (Ours)** | 1.95 ± 0.46 | 34.8 ± 3.0 |

**Computational Efficiency and Wall-Clock Time.** We measure wall-clock training cost on the high-dimensional MuJoCo Playground Humanoid tasks. Diffusion policies incur additional compute due to iterative denoising, and TruDi further adds overhead from evaluating the trust-region constraint against the previous policy at each update. As a result, for a fixed budget of 50M environment steps, TruDi is slower than the Gaussian REPPO baseline and DIME, but this extra cost is offset by substantially better final performance. To ensure a fair comparison against this computational overhead, we additionally evaluate TruDi against REPPO under a strictly matched wall-clock time budget. As shown in Figure 5 (Left), TruDi substantially outperforms REPPO even under identical time constraints ($\approx 2.5$ hours). Furthermore, extending REPPO's training to 150M environment steps (2.78 hours) yields no further improvement in episode return. This indicates that TruDi's advantage stems fundamentally from the expressiveness of the diffusion policy and the stability of our trust-region updates, rather than being an artifact of additional compute time.

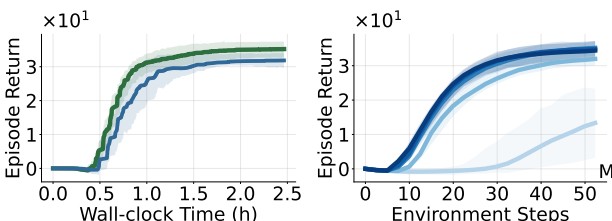

*Figure 5.* **Left**: Wall-clock time comparison on the MuJoCo Playground Humanoid tasks. TruDi substantially outperforms REPPO when evaluated under a strictly matched wall-clock time budget. **Right**: Ablation on diffusion steps $T$. We evaluate $T \in \{1, 4, 8, 16, 32\}$, with colors indicating the step count (e.g., $T = 1$, $T = 4$, $T = 8$, $T = 32$). Performance saturates at $T = 8$.

**Ablation on Diffusion Steps.** To understand the trade-off between performance and computational cost, we ablate the number of diffusion steps $K$ used during training and inference. Figure 5 (Right) illustrates the aggregated performance for $T \in \{1, 4, 8, 16, 32\}$. At $T = 1$, the policy fails to learn effectively, confirming that an iterative diffusion process is essential for representing these complex action distributions. Performance improves steadily as $T$

increases but strongly saturates at $T = 8$. Beyond this threshold, computational time continues to grow linearly without yielding further improvements in final performance, establishing $T = 8$ as the optimal balance between policy expressivity and computational efficiency.

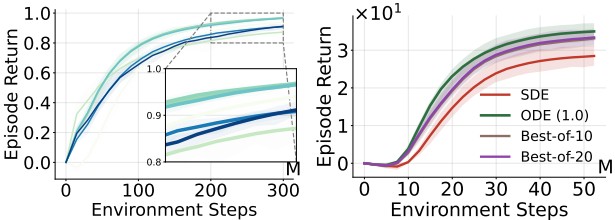

*Figure 6.* **Left**: Sensitivity analysis of the trust-region threshold $\epsilon$. We vary the KL constraint from strict ( 0.01) to loose ( 50) bounds. **Right**: Comparison of policy evaluation strategies, including SDE sampling, ODE sampling, and best-of-$K$ sampling.

**Sensitivity to the trust-region threshold.** We study the effect of the trust-region constraint by varying $\epsilon$ on *IsaacLab benchmark*. We sweep $\epsilon$ from $0.01$ to $50$ and report the aggregated normalized episode return, where curves are color-coded by $\log_{10}(\epsilon)$. As shown in Figure 6 (Left), intermediate values around $0.1$ to $0.4$ (**cyan**) give the best performance, and the optimal region is reasonably broad within this range. When $\epsilon$ is set too large (**dark blue**), the constraint becomes loose, and the update behaves closer to the unconstrained case, which leads to a clear drop in final return. On the other hand, a very small $\epsilon$ is overly conservative, limiting the policy update and resulting in slower learning and lower asymptotic performance.

**Ablation on Evaluation Strategies.** We investigate the impact of different action generation strategies during evaluation. While stochastic sampling (SDE) is essential for exploration during training, deterministic execution with high-return action is more preferred during evaluation. We compare the standard SDE sampler against the Probability Flow ODE (with score scaling $c = 1.0$) and a Best-of-$K$ strategy ($K = \{10, 20\}$), which samples $K$ actions via SDE and selects the one maximizing the learned Q-value. The aggregated results on the MuJoCo Playground Humanoid benchmark (Figure 6 (Right)) demonstrate that the ODE integrator ($c = 1.0$) consistently yields the highest performance. Notably, the ODE solver outperforms the Best-of-$K$ baselines, suggesting that deterministically tracing through probability flow to the mode of the policy distribution is more reliable than the standard sampling strategies.

## 7. Conclusion

We introduced TruDi, a principled approach for training diffusion policies in the on-policy, massively parallel RL regime by enforcing a trust-region constraint over the full diffusion trajectory, which yields a tractable alternative

to marginal KL constraints for diffusion models. Across 4 benchmark suites with 73 tasks, TruDi is competitive with strong Gaussian on-policy baselines on standard control tasks while achieving clear gains on challenging high-dimensional humanoid control, where stable exploration and expressive action distributions matter most. In addition, our analyses highlight practical ingredients for strong performance in this setting, including the sensitivity of the trust-region threshold and the benefit of deterministic policy evaluation via the probability-flow ODE compared to standard SDE sampling and best-of-$K$ selection. Overall, TruDi establishes a strong baseline for combining diffusion policies with trust-region optimization in massively parallel on-policy RL.

**Limitations and Future Work.** A current limitation is the added computational cost of diffusion policies due to iterative sampling, which motivates improving efficiency through fewer denoising steps, distillation, or faster samplers. Looking ahead, it would be interesting to extend our trust-region diffusion framework to more sophisticated training settings, such as offline-to-online RL and sim-to-real transfer with richer observations.

## Impact Statement

This paper presents work whose goal is to advance the field of Robot Learning. Our research enables stable training of expressive diffusion policies in on-policy, massively parallel reinforcement learning, which may improve learning performance on challenging high-dimensional control tasks and broaden the practical use of diffusion models in robotics simulation. As far as we are aware, our work does not raise any specific ethical issues.

## Acknowledgements

We thank the anonymous reviewers for their valuable feedback and suggestions. GN is supported by the European Research Council (ERC) under the European Union's Horizon Europe programme through the project SMARTI³ (Grant Agreement No. 101171393), and by the German Federal Ministry of Research, Technology, and Space (BMFTR) under the Robotics Institute Germany (RIG). The authors acknowledge support by the state of Baden-Württemberg through bwHPC, as well as the HoreKa supercomputer funded by the Ministry of Science, Research and the Arts Baden-Württemberg and by the German Federal Ministry of Education and Research.

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

# A. Derivation of Trajectory KL Upper Bound

The KL divergence between the marginal distributions of the current and old policy to Eq. 14

$$D_{\mathrm{KL}}\left(\bar{\pi}_{\mathrm{old}}(a^0 \mid s) \mid \bar{\pi}_\theta^0(a^0 \mid s)\right) = \int \bar{\pi}_{\mathrm{old}}(a^0 \mid s) \log \frac{\bar{\pi}_{\mathrm{old}}(a^0 \mid s)}{\bar{\pi}_\theta(a^0 \mid s)} da^0 \tag{26}$$

$$= \int \bar{\pi}_{\mathrm{old}}(a^{0:N} \mid s) \log \left( \frac{\bar{\pi}_{\mathrm{old}}(a^{0:N} \mid s)}{\bar{\pi}_\theta(a^{0:N} \mid s)} \frac{\bar{\pi}_\theta(a^{1:N} \mid a^0, s)}{\bar{\pi}_{\mathrm{old}}(a^{1:N} \mid a^0, s)} \right) da^{0:N} \tag{27}$$

$$= D_{\mathrm{KL}}\left( \bar{\pi}_{\mathrm{old}}^{0:N}(a^{0:N} \mid s) \mid\mid \bar{\pi}_\theta(a^{0:N} \mid s) \right) \tag{28}$$

$$- \mathbb{E}_{\bar{\pi}_{\mathrm{old}}(a^0|s)} \left[ D_{\mathrm{KL}}\left( \bar{\pi}_{\mathrm{old}}^{1:N}(a^{1:N} \mid a^0, s) \mid\mid \bar{\pi}_\theta(a^{1:N} \mid a^0, s) \right) \right], \tag{29}$$

where we used the identity $\bar{\pi}^\theta(a^0 \mid s) = \frac{\bar{\pi}^\theta(a^{0:N}|s)}{\bar{\pi}^\theta(a^{1:N}|a^0,s)}$ in Eq. 27. Because the KL is $\geq 0$ we obtain an upper bound. An alternative approach considers applying the data processing inequality (Cover, 1999).

**Remark.** Equation (29) shows that the gap between the true marginal KL (left-hand side) and our upper bound, i.e., the joint trajectory KL (first term on the right-hand side), is precisely given by the conditional KL (second term on the right-hand side). If this term is zero, the bound is tight. However, for stochastic diffusion policies considered in our work, this term is generally nonzero, so tightness is difficult to characterize in full generality. Nevertheless, we empirically showed in Section 6, a valid upper bound on the marginal KL (Equation (28)) is sufficient. This is reflected in Fig. 4 (Left), where overly strict bounds ($\epsilon = 0.01$) lead to slow updates, but moderate values of $\epsilon \in [0.1, 0.4]$ yield strong performance. More broadly, TruDi exhibits robust behavior across a wide range of $\epsilon$ values (Fig. 4), suggesting that the proposed trajectory KL upper bound is effective in practice.

# B. Experiment Details

We evaluate TruDi on four distinct environment suites: **MuJoCo Playground** (Zakka et al., 2025), **HumanoidBench** (Sferrazza et al., 2024), **IsaacLab** (Mittal et al., 2025), and **ManiSkill3** (Tao et al., 2025). These benchmarks were selected to comprehensively test the algorithm's capability across standard continuous control, high-DoF humanoid locomotion, rough-terrain navigation, and contact-rich manipulation.

### B.1. MuJoCo Playground

We utilize the MuJoCo XLA (MJX) implementation provided by MuJoCo Playground (Zakka et al., 2025), which allows for massive parallelization on GPU. We evaluate on two distinct subsets: the classic DeepMind Control (DMC) Suite and the modern Humanoid Joystick tasks.

**DeepMind Control Suite.** We evaluate on a comprehensive set of **20 tasks** covering a wide range of dimensionality and difficulty. The selection includes both dense and sparse reward variants to explicitly test exploration capabilities:

- **Locomotion (7 tasks):** `Cheetah Run`, `Fish Swim`, `Hopper Hop/Stand`, and `Walker Run/Stand/Walk`.

- **Classic Control & Manipulation (13 tasks):** `Acrobot Swingup`, `Ball In Cup`, `Cartpole Balance/Swingup`, `Finger Spin`, `Finger Turn Easy/Hard`, `Pendulum Swingup`, and `Reacher Easy/Hard`.

- **Sparse Variants (included above):** We explicitly include `AcrobotSwingupSparse`, `CartpoleBalanceSparse`, and `CartpoleSwingupSparse` to test exploration in sparse-reward settings.

*Observation & Action Spaces:* The state space $\mathcal{S}$ consists of proprioceptive data (joint positions, velocities) and task-specific features (e.g., target coordinates). The action space $\mathcal{A}$ ranges from $|\mathcal{A}| = 1$ (Pendulum) to $|\mathcal{A}| = 6$ (Walker/Cheetah), normalized to $[-1, 1]$. *Reward:* We use the standard dense rewards defined in the suite, which generally combine a velocity tracking term $\mathcal{R}_{\mathrm{track}}$ with small control regularization penalties $|u|^2$.

**Humanoid Locomotion (G1 & T1).** We evaluate the `Joystick` tasks on `Flat` and `Rough` terrains using two modern humanoid robots: the Unitree G1 (29 DoF) and the Booster T1.

- **Objective:** The agent must track a randomized command vector $c = (v_x, v_y, \omega_z)$ specifying target linear and angular velocities.

- **Observation:** The observation space $\mathcal{S} \in \mathbb{R}^{\approx 60-70}$ includes joint positions and velocities, base linear/angular velocity, the projected gravity vector, and the command history.

- **Reward:** The reward is defined as a product $r_t = r_{\text{tracking}} \cdot r_{\text{penalty}}$. The $r_{\text{tracking}}$ term encourages matching the command velocity, while $r_{\text{penalty}}$ minimizes energy consumption and joint jerk to promote smooth, transferable gaits.

### B.2. HumanoidBench

We evaluate on **HumanoidBench** (Sferrazza et al., 2024), utilizing the `h1hand` variant of the Unitree H1 robot. This setup presents a significantly harder challenge than standard locomotion benchmarks, as it requires the policy to simultaneously manage whole-body balance, locomotion, and fine-grained dexterous manipulation.

**Tasks.** We evaluate on a comprehensive suite of **30 tasks** spanning three distinct behavioral categories. This diverse set tests the agent's ability to generalize across varying dynamic requirements:

- **Locomotion & Agility (13 tasks):** `walk`, `run`, `stand`, `crawl`, `hurdle`, `stair`, `slide`, `maze`, `pole`, `balance_simple/hard`, and `sit_simple/hard`.

- **Whole-Body Manipulation (12 tasks):** `push`, `reach`, `truck`, `package`, `cabinet`, `door`, `window`, `room`, `kitchen`, `basketball`, and `bookshelf_simple/hard`.

- **Dexterous Manipulation (5 tasks):** Fine-motor tasks including `cube`, `spoon`, `powerlift`, and `insert_normal/small`.

**Spaces.** The state space is high-dimensional ($\mathcal{S} \in \mathbb{R}^{\approx 60-90}$), consisting of the root state (orientation, velocity), full-body joint states, and task-specific features such as object poses. Crucially, the action space is $\mathcal{A} \in \mathbb{R}^{61}$, controlling both the 19 actuated joints of the humanoid body and the 42 degrees of freedom of the dexterous hands. This vast action space makes the exploration problem particularly acute, validating the need for the maximum entropy formulation in TruDi.

**Reward.** We utilize the standardized multiplicative reward function provided by the benchmark: $r_t = r_{\text{track}} \cdot r_{\text{energy}} \cdot r_{\text{alive}}$. This structure acts as a soft constraint, ensuring that the agent must maintain stability ($r_{\text{alive}}$) and energy efficiency while maximizing task progress ($r_{\text{track}}$).

### B.3. IsaacLab

We evaluate the robustness and dexterity of learned policies using **IsaacLab** (Mittal et al., 2025), an Omniverse-based simulation framework. Our evaluation covers two distinct domains: rough-terrain humanoid locomotion and high-DoF dexterous in-hand manipulation.

**Humanoid Locomotion.** We evaluate locomotion on four primary tasks utilizing the Unitree G1 and H1 robots: `Isaac-Velocity-Flat-G1-v0`, `Isaac-Velocity-Rough-G1-v0`, `Isaac-Velocity-Flat-H1-v0`, and `Isaac-Velocity-Rough-H1-v0`.

- **Terrain:** While `Flat` tasks operate on a plane, the `Rough` variants utilize procedural terrain generation, presenting slopes, stairs, and discrete obstacles that require robust recovery behaviors.

- **Spaces:** The state space is $\mathcal{S} \in \mathbb{R}^{48}$ for flat terrain. For rough terrain, this expands to $\mathcal{S} \in \mathbb{R}^{235}$ by including a 187-dimensional height-scan for local path planning. The action space consists of continuous PD position targets (23 DoF for G1, 19 DoF for H1).

- **Reward:** We employ a dense reward function tracking linear/angular velocity commands while penalizing joint torques, accelerations, and unstable base orientations.

**Dexterous Manipulation.** We evaluate fine-grained contact control on two in-hand reorientation tasks: `Isaac-Repose-Cube-Allegro-Direct-v0` and `Isaac-Repose-Cube-Shadow-Direct-v0`.

- **Objective:** The agent controls a multi-fingered robotic hand (4-fingered Allegro Hand or 5-fingered Shadow Hand) to rotate a cube to a randomly sampled target orientation.

- **Spaces:** The observation space includes the hand's joint positions and velocities, the object's current pose (position and quaternion), and the target orientation. The action space controls the hand's actuated joints (16 DoF for Allegro, 24 DoF for Shadow Hand), presenting a challenging high-dimensional continuous control problem with frequent contact discontinuities.

### B.4. ManiSkill3

We evaluate fine-grained robotic manipulation capabilities using **ManiSkill3** (Tao et al., 2025), a GPU-parallelized simulator built on the SAPIEN engine (Xiang et al., 2020). Unlike standard rigid-body benchmarks, ManiSkill3 emphasizes contact-rich, physical interaction with diverse object geometries. We evaluate on a custom suite of **13 tasks** utilizing the Franka Emika Panda arm.

**Tasks.** The selected tasks cover a spectrum of manipulation challenges, ranging from basic pick-and-place to high-precision assembly and multi-agent coordination:

- **Standard Manipulation:** `PickCube`, `StackCube`, `PokeCube`, `PullCube`, `PushCube`, and `RollBall`.

- **Precision & Assembly:** `LiftPegUpright`, `PlaceSphere`, `PlugCharger`, and the trajectory-centric `PushT`.

- **Tool Use:** `PullCubeTool`, where the agent must grasp an intermediate tool to manipulate a target object out of reach.

- **Multi-Robot Coordination:** `TwoRobotPickCube` and `TwoRobotStackCube`, requiring the coordination of two independent robot arms (16 DoF total) to solve a shared objective.

**Spaces.** The state space $\mathcal{S} \in \mathbb{R}^{40+}$ generally comprises robot proprioception (joint positions, velocities, gripper width) and ground-truth object states (pose, linear/angular velocities). The action space $\mathcal{A}$ utilizes **delta joint position control** (normalized $\Delta q \in [-1, 1]$), which has been shown to minimize the Sim2Real gap (Tao et al., 2025):

- **Single-Arm:** $\mathcal{A} \in \mathbb{R}^8$ (7 arm joints + 1 gripper).

- **Dual-Arm:** $\mathcal{A} \in \mathbb{R}^{16}$ (concatenated controls for both robots).

**Reward.** We utilize the standardized dense reward functions provided by the benchmark. These typically consist of a *reaching term* (distance to object), a *manipulation term* (distance to target pose), and binary *success indicators* to guide exploration in sparse-contact phases.

### B.5. Baselines

We compare TruDi against the following state-of-the-art baselines, covering standard on-policy methods, and diffusion/flow based policies approaches:

**PPO (Proximal Policy Optimization)** (Schulman et al., 2017): As the standard on-policy algorithm for continuous control, PPO refines the policy gradient objective by clipping the probability ratio to a trust region, ensuring monotonic improvement. We use the highly optimized implementation from RSL RL (Schwarke et al., 2025), parameterizing the policy as a diagonal Gaussian distribution where the mean and standard deviation are output by the actor network. Value targets are computed using Generalized Advantage Estimation (GAE), and the network is trained via the Adam optimizer.

**SPO (Simple Policy Optimization)** (Xie et al., 2025): This novel unconstrained first-order algorithm is designed to enforce trust region constraints more effectively than PPO's hard clipping. SPO employs a specialized objective that imposes a soft constraint on the probability ratio deviation, dynamically scaled by the magnitude of the advantage estimates. Specifically,

the policy loss is formulated as $\mathbb{E}[r_t(\theta)\hat{A}_t - \frac{|\hat{A}_t|}{2\epsilon}(r_t(\theta) - 1)^2]$, which penalizes large deviations from the behavior policy without zeroing out gradients (unlike clipping). We utilize the official implementation and adapt it on top of the RSL RL PPO framework with GAE and Adam, using an MLP-based diagonal Gaussian parameterization to ensure a fair comparison with the PPO baseline.

**FPO (Flow Policy Optimization)** (McAllister et al., 2025): An on-policy formulation that integrates flow matching into the policy gradient framework. FPO circumvents the intractability of exact likelihood computation in flow-based models by casting policy optimization as maximizing an advantage-weighted ratio derived from the conditional flow matching loss. This objective is optimized within a PPO-style clipping framework. We use the official implementation, parameterizing the vector field with an MLP.

**DPPO (Diffusion Policy Policy Optimization)** (Ren et al., 2024): A systematic framework for fine-tuning diffusion-based policies using on-policy RL. DPPO adapts the PPO objective to diffusion models by parameterizing the policy as a conditional diffusion model at each denoising step. To enable efficient online training, we utilize the official codebase and implement it top of the RSL RL PPO framework, parameterizing the diffusion backbone as an MLP with sinusoidal timestep embeddings. The policy is optimized using Adam with a learning rate decay schedule, while the critic remains a standard MLP trained to minimize temporal difference error.

**REPPO (Relative Entropy Pathwise Policy Optimization)** (Voelcker et al., 2025): An on-policy algorithm that enables the use of low-variance pathwise gradients (reparameterization trick) by training a critic on on-policy data, stabilized via a relative entropy (KL) trust-region constraint. This serves as a critical ablation: by comparing Gaussian-REPPO against our diffusion-based method, we isolate the performance gains attributable to the expressivity of the generative parameterization from those derived from the underlying pathwise optimization objective.

**DIME (Diffusion-Based Maximum Entropy RL)** (Celik et al., 2025): The state-of-the-art off-policy algorithm for diffusion policies. DIME extends the standard MaxEnt framework to diffusion models by deriving a tractable lower bound on the policy entropy, allowing for principled exploration without ad-hoc regularizers. We utilize identical MLP backbones for DIME as in our proposed method to remove confounding factors related to network architecture.

## C. Implementation and Training Details

We provide a comprehensive overview of the network architectures, training procedures, and hyperparameters used for TruDi. Our implementation leverages both the JAX (Bradbury et al., 2018) and PyTorch (Paszke et al., 2019) frameworks to maximize compatibility and training throughput.

**Network Architectures.** The diffusion policy $\pi_\theta(a|s)$ is parameterized as a conditional noise prediction network $\epsilon_\theta(x_t, t, s)$. Following the design principles in DIME (Celik et al., 2025), we utilize a Multi-Layer Perceptron (MLP) adapted for low-dimensional state-action spaces, departing from the U-Net architectures typically used in image generation. The network conditions on the state $s$ and the diffusion timestep $t$. We encode $t$ using sinusoidal Fourier features which is the same as in (Vaswani et al., 2017) followed by a 2-layer MLP projection. The state $s$, noisy action $x_t$, and time embedding are concatenated and passed through a residual backbone consisting of **3 hidden layers** with 512 units each and **GeLU** activations. We apply Layer Normalization (Ba et al., 2016) to the inputs of each residual block to stabilize training deep diffusion priors.

For the value function, we adopt the specific architecture proposed in REPPO (Voelcker et al., 2025) to ensure stable pathwise gradient propagation. The critic $Q_\phi(s, a)$ is parameterized as a 3-layer MLP (2 hidden layers) with 512 hidden units and ReLU activations. Crucially, we apply Layer Normalization (Ba et al., 2016) to the first hidden layer of the critic. This normalization prevents the scale of the value gradients from diverging, which is essential for maintaining the trust region without aggressive gradient clipping.

For the value function, we **follow the design proposed in REPPO** (Voelcker et al., 2025) to ensure stable pathwise gradient propagation. The critic $Q_\phi(s, a)$ is parameterized as a **3-layer MLP** with 512 hidden units and **SiLU** activations. To accurately model the value uncertainty, we employ a distributional head with a support size of **151 bins**. Crucially, consistent with the REPPO architecture, we apply Layer Normalization to the first hidden layer of the critic to prevent value gradient scaling issues.

**Training Procedure.** Training proceeds in an on-policy fashion. In each iteration, we collect trajectories by rolling out the current diffusion policy in parallel environments. Actions are generated using the DIME scheduler (Celik et al., 2025) with $K = 8$ denoising steps. We normalize environment observations using online running statistics (mean and variance), which are updated during rollouts and frozen during gradient updates.

Following the training recipe of REPPO (Voelcker et al., 2025), we utilize TD($\lambda$) (Sutton, 1988) to compute targets for the critic, rather than Generalized Advantage Estimation (GAE). Specifically, we calculate the entropy-augmented $\lambda$-return $G_t^\lambda$ to serve as the regression target for the value function. This formulation effectively balances the bias-variance tradeoff in the target estimates without requiring a separate advantage computation. The policy is then updated to maximize the learned Q-value via pathwise gradients, subject to a trust-region constraint where the intractable entropy term is approximated via the diffusion training loss.

**Hyperparameters and Resources.** We use the Adam optimizer for both the actor and critic networks. Table 3 summarizes the core hyperparameters used across the MuJoCo Playground and HumanoidBench experiments. All experiments were conducted on a high-performance computing cluster equipped with NVIDIA A100 (80GB) and H100 GPUs.

*Table 3.* **Hyperparameters for TruDi.**

| Parameter | Value |
|---|---|
| *General Settings* | |
| Total Timesteps (Locomotion) | $5 \times 10^7 - 3 \times 10^8$ |
| Total Timesteps (Manipulation) | $5 \times 10^7 - 3 \times 10^8$ |
| Num Environments | $2048 - 4096$ |
| Rollout Length | 128 |
| Discount Factor $\gamma$ | 0.99 |
| GAE $\lambda$ | 0.95 |
| *Diffusion Policy (Actor)* | |
| Architecture | Residual MLP + LayerNorm |
| Hidden Layers | 3 |
| Hidden Dimension | 512 |
| Time Dimension | 32 |
| Activation | GeLU |
| Diffusion Steps $T$ | 8 |
| Noise Schedule | Cosine ($\beta_{min} = 10^{-4}, \beta_{max} = 2 \times 10^{-2}$) |
| *Value Function (Critic)* | |
| Architecture | MLP + LayerNorm |
| Hidden Layers | 3 |
| Hidden Dimension | 512 |
| Activation | GeLU |
| Ensemble/Bin Size | 256 |
| *Optimization* | |
| Optimizer | Adam |
| Actor Learning Rate | $3 \times 10^{-4}$ |
| Critic Learning Rate | $3 \times 10^{-4}$ |
| Mini-batch Size | 2048 |
| Num Epochs | 8 |
| Trust Region $\epsilon$ | 0.1 |

# D. Multi-Modal Validation Tasks

To highlight the multi-modal solution generation capabilities of TruDi, we design two illustrative tasks: **Push-T** and **StackCube**. These tasks are specifically engineered to possess symmetric solutions, allowing us to test whether the policy can learn and represent multiple distinct geometric modes simultaneously.

### D.1. Push-T Task

In the Push-T task, the robot must rotate a T-shaped block $\pi$ radians (180 degrees) to match a goal orientation. The task is designed with inherent geometric symmetry: from a neutral starting position, rotating the block either clockwise or counter-clockwise constitutes a valid and optimal solution. A uni-modal policy (e.g., Gaussian) often averages these modes, resulting in failure (e.g., getting stuck in the middle), whereas a multi-modal policy should be able to commit to one specific direction.

#### D.1.1. REWARD DESIGN

We implement these environments as modified versions of the standard Push-T task from the ManiSkill simulation framework (Tao et al., 2025), specifically adapted to highlight multi-modal solution capabilities.

**PushT-Train (Training Environment).**    To ensure robust policy learning, the training environment features full randomization. Both the T-shaped block and the goal position are initialized randomly within the workspace bounds ($x \in [-0.3, 0.1]$m, $y \in [-0.3, 0.2]$m), with random Z-axis rotations $\theta \in [0, 2\pi]$.

We employ a dense, shaped reward function $r = r_{\text{rot}} + r_{\text{pos}} + r_{\text{tcp}}$ (max value 3.0), consisting of three components:

1. **Rotation Alignment:**

$$r_{\text{rot}} = \frac{1}{2} \left( \frac{\cos(\theta_{\text{block}} - \theta_{\text{goal}}) + 1}{2} \right)^2 \tag{30}$$

   where $\theta$ denotes the Z-axis rotation in radians. The squaring operation amplifies rewards as the alignment approaches perfection.

2. **Position Alignment:**

$$r_{\text{pos}} = \frac{1}{2} \left( 1 - \tanh(5\|p_{\text{block}} - p_{\text{goal}}\|_2) \right)^2 \tag{31}$$

   where $p \in \mathbb{R}^2$ denotes the 2D position.

3. **Manipulation Guidance:**

$$r_{\text{tcp}} = \frac{1}{20} \sqrt{1 - \tanh(5\|p_{\text{tcp}} - p_{\text{block}}\|_2)} \tag{32}$$

   This small shaping term encourages the tool center point (TCP) to remain close to the block.

**Success Criterion:** Success is defined as $\geq 90\%$ intersection area between the block and goal footprints.

**PushT-Test (Multi-Modal Evaluation).**    We design a deterministic test environment to isolate multi-modal decision-making. The goal is fixed at the workspace center with zero rotation, and the block is initialized at the same position but rotated exactly $\pi$ radians. Crucially, the reward function is **rotation-symmetric**: rotating by $+\theta$ (clockwise) or $-\theta$ (counter-clockwise) yields identical rewards. This symmetry enables us to quantitatively analyze whether TruDi can model the bimodal distribution of valid trajectories.

### D.2. StackCube Task

To further evaluate multi-modality in a contact-rich manipulation setting, we extend the standard stacking task. In our setup, two cubes (Red and Blue) are placed in the scene. The goal is simply to stack *one* cube on top of the *other*. This creates two valid geometric modes: **Red-on-Blue** or **Blue-on-Red**.

#### D.2.1. REWARD DESIGN

We implement this task by extending the ManiSkill (Tao et al., 2025) stacking environment.

**StackCube-Train (Training Environment).**    During training, the environment is fully randomized to promote generalization. The Red cube (Cube A) and Blue cube (Cube B) spawn at random XY positions, and the specific stacking goal (A-on-B or B-on-A) is randomly selected per episode. The dense reward function consists of five stages (max value 8.0):

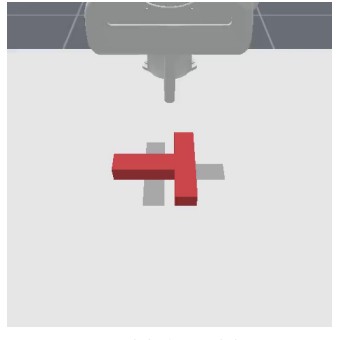
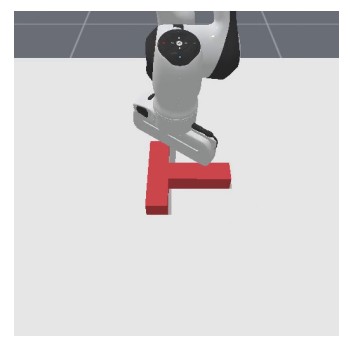
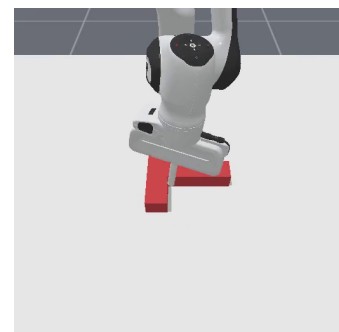

(a) Initial Position        (b) Mode 1: Clockwise        (c) Mode 2: Counter-Clockwise

*Figure 7.* **Push-T Multi-Modal Task.** The agent must rotate the T-block $\pi$ radians. The task admits two symmetric solutions (modes), requiring the policy to commit to one direction rather than averaging them.

1. **Reaching (Stage 1):** Guides the end-effector to the nearest cube:

$$r_{\text{reach}} = 2\left(1 - \tanh(5d_{\text{tcp-nearest}})\right) \tag{33}$$

   where $d_{\text{tcp-nearest}} = \min(\|p_{\text{tcp}} - p_{\text{A}}\|, \|p_{\text{tcp}} - p_{\text{B}}\|)$.

2. **Placing (Stage 2):** Uses a symmetric maximum operator to avoid biasing the policy toward a specific cube order:

$$r_{\text{place}} = \max(r_{\text{A-on-B}}, r_{\text{B-on-A}}) \tag{34}$$

   where $r_{\text{X-on-Y}}$ encourages moving cube X to the top of cube Y.

3. **Release & Stability (Stage 3):** Once stacked, additional terms $r_{\text{ungrasp}}$ and $r_{\text{static}}$ encourage the robot to release the object and ensure the stack is stable.

**StackCube-Test (Multi-Modal Evaluation).** The testing environment is deterministic and explicitly constructed to challenge the policy's mode-selection capability.

- **Setup:** Cube A (Red) is placed at the far left ($y = -0.35$m) and Cube B (Blue) at the far right ($y = +0.35$m).

- **Challenge:** The wide lateral separation (0.7m) creates two disjoint basins of attraction in joint space. A valid policy must essentially "choose" between a leftward trajectory (stack Red-on-Blue) or a rightward trajectory (stack Blue-on-Red).

- **Symmetry:** The reward function remains identical to the training setup. The $\min$ operator in reaching and the $\max$ operator in placing ensure the reward landscape is perfectly symmetric, allowing the agent to dynamically select either mode.

### D.3. Evaluation Metric: Behavior Entropy

To quantitatively evaluate the policy's ability to cover both symmetric solutions, we adopt the **Behavior Entropy** metric adapted from the D3IL benchmark (Jia et al., 2024). This metric measures the diversity of the behaviors by computing the entropy of the discrete mode distribution achieved by the policy.

For a set of $N$ evaluation trajectories, we classify each trajectory $\tau_i$ into a binary mode $m \in \{0, 1\}$ using a task-specific classifier $C(\tau_i)$. We estimate the empirical probability of each mode as $\hat{p}_m = \frac{1}{N} \sum_{i=1}^{N} \mathbb{I}(C(\tau_i) = m)$.

Since our tasks contain exactly two distinct modes ($|\mathcal{M}| = 2$), we calculate the entropy using the base-2 logarithm:

$$\mathcal{H}_{\text{behavior}} = -\sum_{m \in \{0,1\}} \hat{p}_m \log_2(\hat{p}_m) \tag{35}$$

This formulation yields an entropy score in the range $[0, 1]$. A score of $\mathcal{H}_{\text{behavior}} = 1$ indicates maximum diversity (a perfect 50/50 split between modes), while $\mathcal{H}_{\text{behavior}} = 0$ indicates mode collapse (the policy executes only one solution).

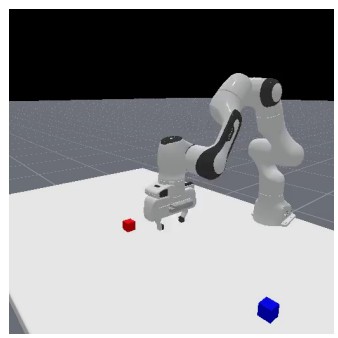
(a) Initial Position

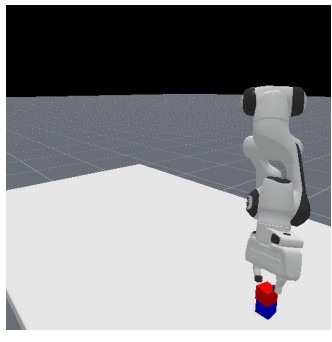
(b) Mode 1: Red on Blue

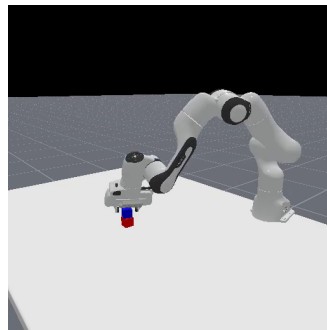
(c) Mode 2: Blue on Red

*Figure 8.* **StackCube Multi-Modal Task.** Two cubes are placed far apart. The agent must choose to either stack Blue on Red (Right-to-Left) or Red on Blue (Left-to-Right). Gaussian policies often fail by averaging these distinct trajectories.

**Mode Definitions.**    We classify the mode of a trajectory based solely on the environment state at the final timestep. The specific criteria for our tasks are:

- **Push-T** ($|\mathcal{M}| = 2$)**:** Modes are distinguished by the sign of the total Z-axis rotation $\Delta\theta$ (derived from the block's final quaternion):

    - *Mode 0 (Counter-Clockwise):* $\Delta\theta > 0$.
    - *Mode 1 (Clockwise):* $\Delta\theta < 0$.

- **StackCube** ($|\mathcal{M}| = 2$)**:** Modes are distinguished by the final vertical arrangement of the cubes (where $h_{\text{cube}} = 0.04$ m):

    - *Mode 0 (Blue-on-Red):* $z_{\text{blue}} > z_{\text{red}} + h_{\text{cube}}$.
    - *Mode 1 (Red-on-Blue):* $z_{\text{red}} > z_{\text{blue}} + h_{\text{cube}}$.

# E. TruDi Main Experiments

## E.1. Hyperparameters

Table 4 lists the default hyperparameters used for the algorithms. We maintain a consistent architecture for generative policies while adapting sampling budgets to the specific requirements of each environment (Table 5). Note that for the Humanoid-Bench tasks, TruDi and REPPO use a reduced hidden dimension of 256, while SPO, PPO, and DPPO use a reduced hidden dimension of 128.

*Table 4.* Algorithm Hyperparameters. Constant across all experiments unless noted.

| Parameter | TruDi (Ours) | REPPO | SPO | PPO | FPO | DPPO |
|---|---|---|---|---|---|---|
| *Actor Network* | | | | | | |
| Hidden Dim | 512 | 512 | 256 | 256 | 32 | 256 |
| Hidden Layers | 3 | 3 | 3 | 3 | 5 | 3 |
| Activation | GeLU | GeLU | ELU | ELU | SiLU | Mish |
| Flow/Diff Steps | 8 | N/A | N/A | N/A | 10 | 8 |
| *Critic Network* | | | | | | |
| Critic Type | Dist. (HL-Gauss) | Dist. (HL-Gauss) | MSE | MSE | MSE | MSE |
| Hidden Dim | 512 | 512 | 256 | 256 | 256 | 256 |
| Hidden Layers | 3 | 3 | 3 | 3 | 5 | 3 |
| Ensemble/Bin | 151 Bins | 151 Bins | 1 | 1 | 1 | 1 |
| Activation | GeLU | GeLU | ELU | ELU | SiLU | Mish |
| *Optimization* | | | | | | |
| Optimizer | | | Adam | | | |
| Actor LR | $3 \times 10^{-4}$ | $3 \times 10^{-4}$ | $1 \times 10^{-3}$ | $1 \times 10^{-3}$ | $3 \times 10^{-4}$ | $3 \times 10^{-4}$ |
| Critic LR | $3 \times 10^{-4}$ | $3 \times 10^{-4}$ | $1 \times 10^{-3}$ | $1 \times 10^{-3}$ | $3 \times 10^{-4}$ | $3 \times 10^{-4}$ |
| Max Grad Norm | 0.5 | 0.5 | 1.0 | 1.0 | 1.0 | 1.0 |
| Constraints | $\epsilon = 0.1$ (KL) | $\epsilon = 0.1$ (KL) | $\epsilon = 0.2$ (Penalty) | $\epsilon = 0.2$ (Clip) | 0.2 (Clip) | 0.2 (Clip) |

*Table 5.* Task-Specific Configurations. Summary of environment settings. We align the sampling budget (Envs $\times$ Horizon) and discount factors ($\gamma$) to the specific needs of each benchmark.

| Benchmark | Total Steps | $\gamma$ | Critic Range | Sampling (Envs $\times$ Horizon) |
|---|---|---|---|---|
| MuJoCo (DMC) | 50M | 0.99 | $[0, 150]$ | $1024 \times 128$ |
| MuJoCo (Humanoid) | 50M | 0.97 | $\pm 10$ | $1024 \times 128$ |
| IsaacLab | 300M | 0.97 | $\pm 10$ | $4096 \times 64$ |
| ManiSkill3 | 50M | 0.99 | $\pm 15$ | $1024 \times 128$ |
| HumanoidBench | 50M | 0.99 | $\pm 250$ | $128 \times 128$ |

# F. Per-Task Learning Curves

In this section, we provide sample efficiency curves per environment.

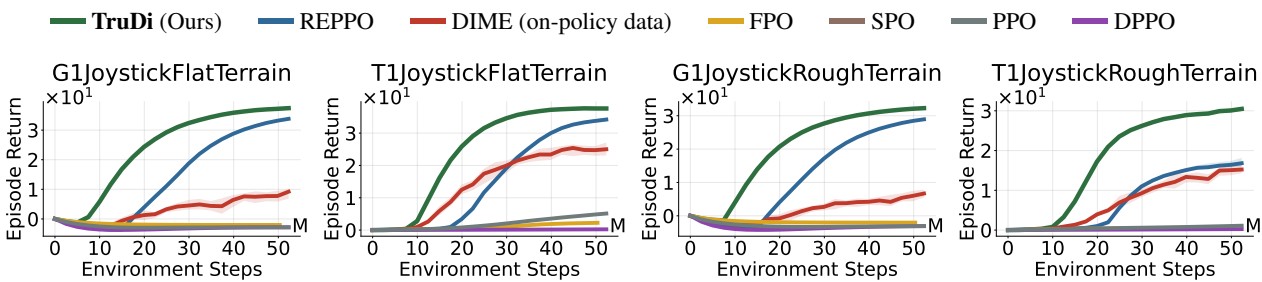

*Figure 9.* IQM Episode Return of each individual Mujoco Playground Humanoid Tasks.

*Figure 10.* IQM Episode Return of each individual Mujoco Playground DMC tasks.

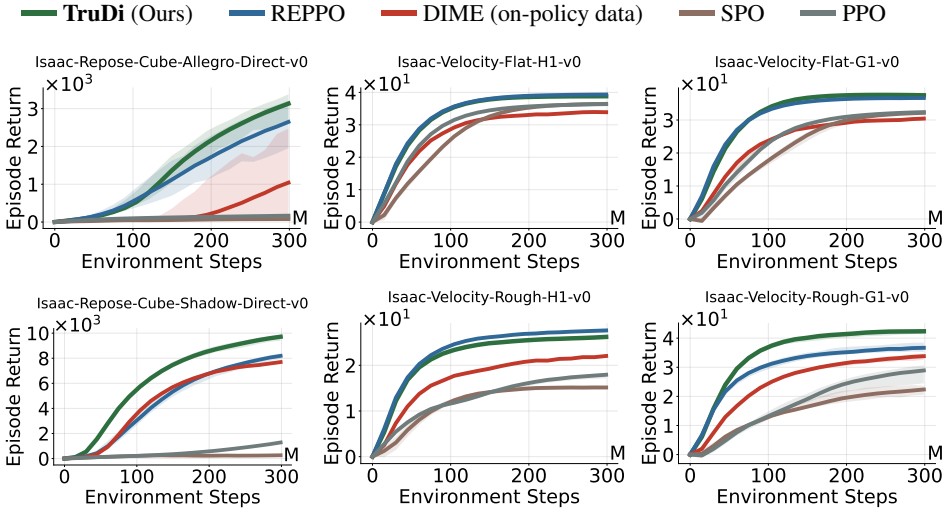

*Figure 11.* IQM Episode Return of each individual ManiSkill tasks.

*Figure 12.* IQM Episode Return of each individual IsaacLab Tasks.

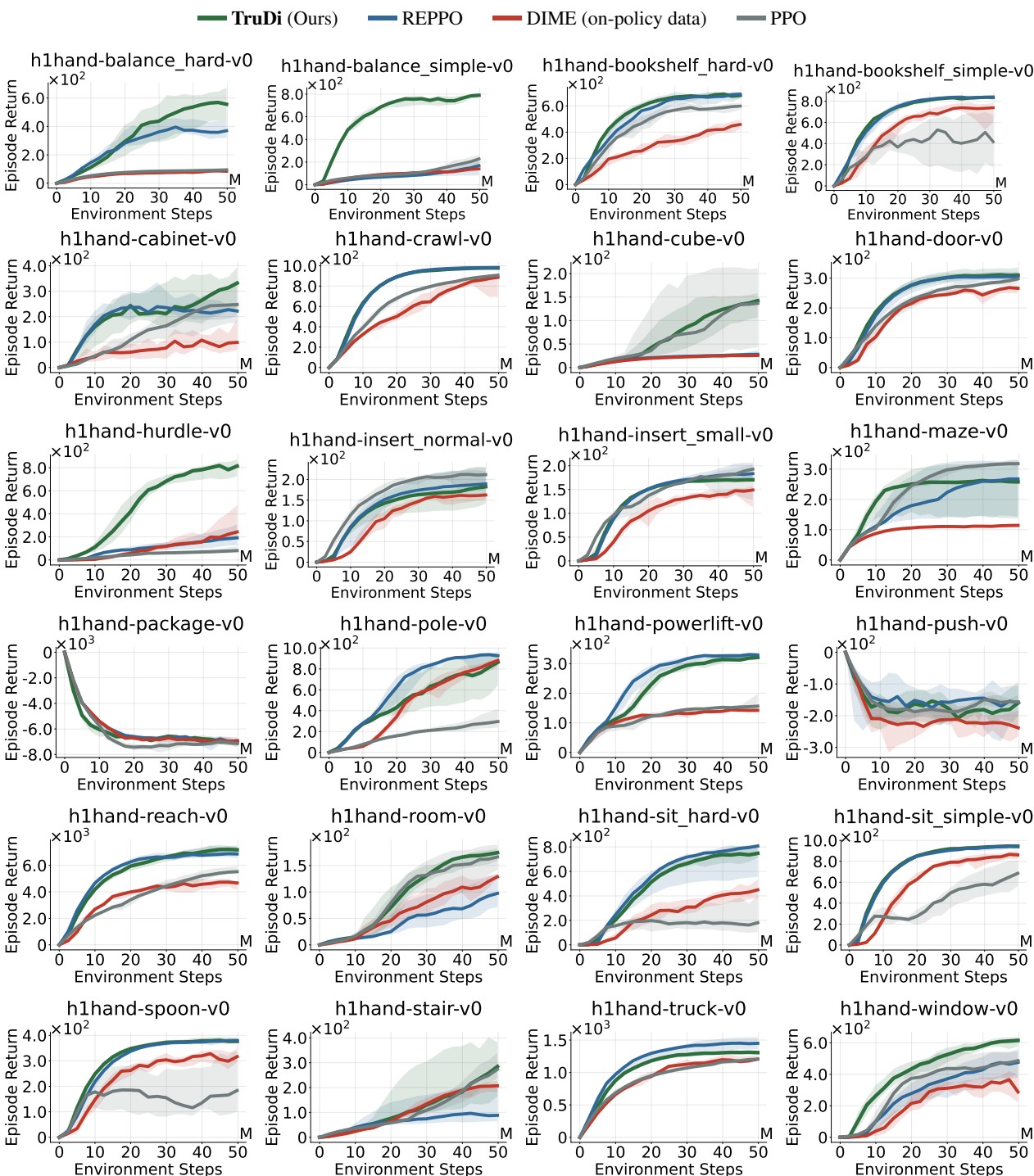

*Figure 13.* IQM Episode Return of each individual Humanoid-Bench Tasks (Part 1).

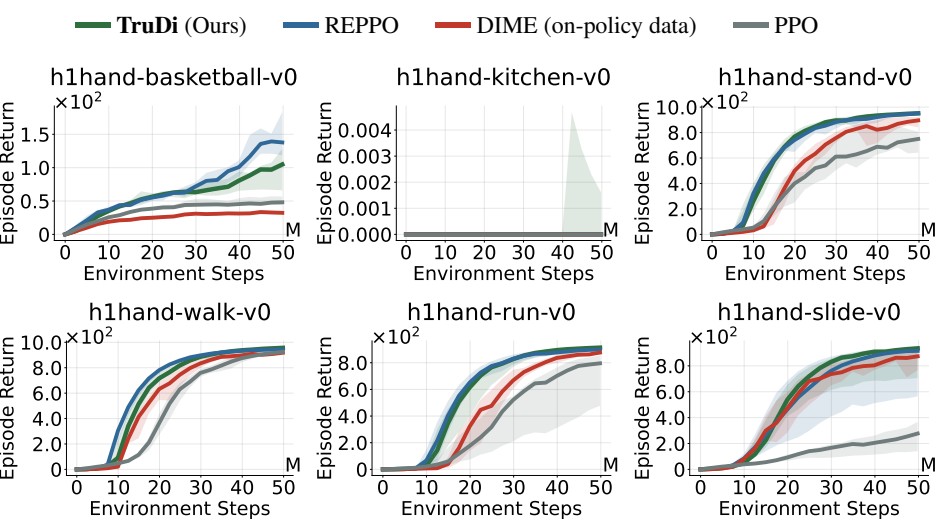

*Figure 14.* IQM Episode Return of each individual Humanoid-Bench Tasks (Part 2).

