# OpenReview forum: "Trust-Region Diffusion Policies for Massively Parallel On-Policy RL"
_ICML.cc/2026/Conference — ICML 2026 regular_

### Official Review · Reviewer_2hoS · 2026-02-27

**Soundness:** 2
**Presentation:** 1
**Significance:** 2
**Originality:** 1
**Overall Recommendation:** 4
**Confidence:** 2

**Summary:**

This paper introduces Trust-region Diffusion Policies (TruDi), which trains diffusion policies in an on-policy manner using massively parallel environments. To enable stable training, TruDi enforces a trust region for diffusion policies by imposing a KL-divergence constraint over the entire diffusion trajectory. Experiments on manipulation and locomotion tasks show that TruDi outperforms several baselines.

**Compliance With Llm Reviewing Policy:**

Affirmed.

**Final Justification:**

My concerns have been adequately addressed, and sorry for the significant misunderstanding due to an initial reading. After a careful study of the formulas and their derivations in the method section, as well as the comments from other reviewers, I was finally able to grasp the core ideas of the paper. I will raise my score to 4 and adjust my confidence to 2 accordingly.

**Key Questions For Authors:**

See Weaknesses above.

**Limitations:**

Yes.

**Strengths And Weaknesses:**

Strengths:
- The authors conduct extensive experiments to validate the effectiveness of the proposed method.
- The motivation is reasonable: since diffusion models do not provide tractable action densities, training diffusion policies under SAC-style paradigms remains largely unexplored.

Weaknesses:
- This paper is poorly written with limited contribution.
- The method section is confusing, with a large number of equations presented with little explanation. The authors should better articulate their core contributions and revise the manuscript to be more logically organized and easier to follow.

Overall, the paper is not ready for publication.

---

> ### Author Rebuttal · Authors · 2026-03-30
>
> We sincerely thank the reviewer for taking the time to review our paper and sharing their concerns about the core contributions as well as the overall writing quality of the paper. However, we would like to recall here that the other three reviewers found the paper's contributions to be substantial, describing the theoretical formulation as "principled" and "elegant" (auMA), the technical idea as "interesting" (F4A3) with a "neat way to bypass" the intractable likelihood problem (KToi), and the experimental evaluation as "comprehensive" (auMA) and "extensive" (F4A3).
>
> Nevertheless, in the following paragraphs, we will walk through the core problem we tackle, our contributions, and the paper's structure, hoping this helps clarify the nature and scope of our work. We also welcome any follow-up suggestions on which parts could be improved further.
>
> **Problem statement.** We want to study whether using a more expressive policy representation, such as diffusion-based policies, is beneficial in the massively parallel on-policy RL setting.
>
> However, due to the high-frequency data collection manner, on-policy methods naturally require estimating trust-region constraints to stabilize the training process. For Gaussian policies this is straightforward, as the density can be obtained using a closed-form formula, but for diffusion policies, the marginal likelihood requires integrating over all latent denoising steps, which is intractable.
>
> We provide here a short walkthrough of our solutions, which form our main contributions:
>
> 1. First, following the latent variable model perspective on diffusion policies from the prior work (DIME [4]), we interpret the denoising chain as a latent variable model and apply the data processing inequality to obtain a tractable lower bound on the entropy (Eq. 12).
> 2. **Next, we provide a tractable trust-region constraint for diffusion policies (Section 4.2).** This is the core technical contribution. Here, we show that the intractable marginal KL divergence (Eq. 9) can be upper-bounded by the KL between the full diffusion trajectories of the old and new policies (Eq. 14). Because the trajectory distributions factorize into Gaussian transitions (Eq. 7), this upper bound can be estimated efficiently from samples. Constraining this bound gives us a more principled way to approximate the trust region for diffusion policies.
> 3. Beyond the training objective, we also identify the stochastic sampling method inherent to diffusion as a major bottleneck for policy evaluation. In RL, it is common to report the quality of a learned policy after removing exploration noise. While this is straightforward for Gaussian policies, it is not obvious for diffusion models. **We propose to use the ODE evaluation as  a noise-reduced variant of sampling from the diffusion policy (Section 4.3)**. This enables a more direct and fair comparison between diffusion and Gaussian policies in an exploration-noise-free setup. If we instead followed the standard stochastic evaluation protocol for diffusion policies, as done in prior work [4, 5, 6], the benefits of using diffusion models would be harder to observe. Please also see the SDE vs. ODE ablation in our response to Reviewer KToi (Q3).
> 4. **Finally, we evaluate our method on a comprehensive set of tasks.** To train our diffusion policy in the on-policy RL setting, we follow REPPO algorithm to compute low-variance pathwise gradients through a learned Q-function. In the empirical evaluation section, our method, named TruDi, demonstrates strong performance across **73 tasks in** **4 benchmark** suites over **10 seeds**.
>
> We hope this short walkthrough helps clarify the contributions and structure of our work. We would be very grateful for any specific pointers on which equations or passages were most difficult to follow, as this would help us target our revisions effectively. We welcome any further discussion and are glad to address additional questions.
>
> [4] Celik et al., 2025. DIME: Diffusion-based maximum entropy reinforcement learning.
>
> [5] Ding et al., 2024. Diffusion-based Reinforcement Learning via Q-weighted Variational Policy Optimization.
>
> [6] Le et al., 2025. Enhancing Exploration With Diffusion Policies in Hybrid Off-Policy RL: Application to Non-Prehensile Manipulation.

---

> > ### Author Rebuttal · Reviewer_2hoS · 2026-04-02
> >
> > My concerns have been adequately addressed, and sorry for the significant misunderstanding due to an initial reading. After a careful study of the formulas and their derivations in the method section, as well as the comments from other reviewers, I was finally able to grasp the core ideas of the paper. I will raise my score to 4 and adjust my confidence to 2 accordingly.

---

> > > ### Author Response · Authors · 2026-04-07
> > >
> > > We sincerely thank the reviewer for revisiting the paper and updating the score. We will improve the presentation in the camera-ready version by adding more intuition and clearer guidance around the key derivations.

---

### Official Review · Reviewer_auMA · 2026-03-09

**Soundness:** 3
**Presentation:** 3
**Significance:** 3
**Originality:** 3
**Overall Recommendation:** 5
**Confidence:** 4

**Summary:**

This paper proposes Trust-region Diffusion Policies (TruDi), a method for training diffusion-based policies in the on-policy, massively parallel reinforcement learning setting. The key challenge is that intractable marginal likelihoods of diffusion models prevent direct application of standard trust-region constraints. TruDi views the diffusion process as a latent variable model and derives a tractable upper bound on the marginal KL divergence via the KL divergence over the full diffusion trajectory. Additionally, the paper proposes using the probability-flow ODE with a scaled score function for deterministic policy evaluation. Experiments span 4 massively parallel RL benchmark suites covering 73 tasks. Results show TruDi is competitive with strong Gaussian baselines on standard tasks and achieves clear gains on high-dimensional humanoid control tasks.

**Compliance With Llm Reviewing Policy:**

Affirmed.

**Final Justification:**

The rebuttal addressed majority of my concerns. While some limitation still exists, I believe this paper has satisfied the requirements of ICML and opens a valuable direction for future research. Therefore, I think this paper may be accepted.

**Key Questions For Authors:**

1. **Bound tightness:** Can you empirically compare trajectory-level KL with marginal KL?

2. **Computational efficiency:** Can you decompose overhead into denoising vs. trust region? How does performance scale with K?

3. **Standard task performance:** Is the diffusion policy advantage limited to high-dimensional multimodal tasks?

4. **Dual updates:** What is the empirical constraint violation profile during training?

5. **Off-policy comparison:** How does DIME perform in its native off-policy setting on the same benchmarks?

**Limitations:**

Briefly discusses computational cost and mitigations. Missing: (1) bound approximation conservatism; (2) limited advantage on simple tasks; (3) dual update constraint violation impact. Societal impact discussion adequate.

**Strengths And Weaknesses:**

**Strengths:**

1. **Soundness - Principled theoretical formulation:** The latent variable model perspective with data processing inequality to derive the trajectory KL upper bound is elegant and correct. More principled than heuristic approaches like DPPO's per-step clipping.

2. **Soundness - Comprehensive experiments:** 4 benchmark suites, 73 tasks, IQM with 95% CI over 10 seeds. Thorough baseline selection covering both Gaussian and diffusion/flow-based methods.

3. **Significance - Addresses a meaningful gap:** First principled solution for training diffusion policies in on-policy massively parallel RL.

4. **Presentation - Clear structure with valuable ablations:** Sensitivity analysis on epsilon and evaluation strategy comparison provide practical guidance. Multimodality experiments cleverly demonstrate diffusion policy expressiveness.

5. **Originality - Score scaling for evaluation:** Simple yet effective temperature parameter for controlling greediness during evaluation.

**Weaknesses:**

1. **Soundness - Bound tightness not analyzed:** The trajectory KL upper bound's tightness is never examined. If excessively loose, it may cause overly conservative updates.

2. **Soundness - Computational overhead understated:** 82% increase in wall-clock time (1.95h vs 1.07h) is not "marginal." Only reported for one benchmark.

3. **Significance - Limited advantage on standard tasks:** Minimal margin over REPPO on DMC, ManiSkill, IsaacLab. Primary advantage confined to humanoid tasks.

4. **Originality - Incremental combination:** Combines DIME objective + REPPO framework + latent variable perspective + DPI bound. Each component is established.

5. **Soundness - No convergence guarantees for dual updates:** Constraint violation frequency not analyzed.

6. **Presentation - Insufficient off-policy comparison:** DIME not evaluated in its native off-policy setting.

7. **Soundness - Missing diffusion steps ablation:** K=8 fixed without analyzing impact of varying K.

---

> ### Author Rebuttal · Authors · 2026-03-30
>
> We thank Reviewer auMA for the detailed and constructive review, and for recognizing the principled formulation and comprehensive experiments.
>
> W1 (Bound tightness - Q1). Please see our response to Reviewer F4A3 (W1) for a detailed discussion.
>
> W2 (Computational overhead - Q2). We thank the reviewer for this comment and agree that this does not look like a marginal overhead at first glance. We reported wall-clock time on the Humanoid tasks specifically because they are the hardest benchmark in our evaluation. Even there, TruDi outperforms REPPO under the same wall-clock time budget, and REPPO cannot reach TruDi's performance even with 3x more training steps (see our response to Reviewer KToi, Q1). Additionally, the computational overhead is dominated by the denoising steps (K=8 forward passes vs. 1 for Gaussian), not the trust-region computation itself. Our K-ablation (W7 below) shows that K=4 already achieves 93% of K=8 performance at only 48% overhead (1.58h vs. 1.07h).
>
> W3 (Limited advantage on standard tasks - Q3). TruDi does outperform the baselines on standard and manipulation tasks as well, though the margin is smaller than on humanoid tasks. This is expected: on simpler tasks where a Gaussian policy is already sufficient, there is less room for a more expressive policy to improve. The advantage of diffusion policies becomes more pronounced precisely on the harder tasks that require richer action distributions.
>
> W4 (Incremental combination). We respectfully disagree that the contribution is merely incremental. While individual components are established, the key insight is that the trajectory KL upper bound (Eq. 14) provides a principled bridge between the DIME maximum entropy objective (designed for off-policy) and the trust-region constraints (essential for on-policy stability). This bridge is non-trivial: prior work (DPPO, GenPO) required either heuristic approximations or action-space modifications. Our formulation is clean, requires no architectural changes, and provides a principled trust-region framework for training diffusion policies in massively parallel on-policy RL.
>
> W5 (Convergence guarantees - Q4). Dual descent without formal convergence guarantees is standard practice in deep RL (SAC, REPPO). We provide empirical evidence on the 20 MuJoCo Playground DMC tasks (with $\epsilon = 0.1$) showing that the dual variables behave well throughout training:
>
> | Step (M) | train/kl (mean ± std) | train/lagrangian (mean ± std) |
> | --- | --- | --- |
> | 2.5 | 0.0934 $\pm$ 0.0331 | 0.0485 $\pm$ 0.0444 |
> | 10.0 | 0.0959 $\pm$ 0.0299 | 0.0148 $\pm$ 0.0131 |
> | 19.9 | 0.0980 $\pm$ 0.0901 | 0.0077 $\pm$ 0.0080 |
> | 29.9 | 0.0918 $\pm$ 0.0297 | 0.0060 $\pm$ 0.0069 |
> | 39.8 | 0.0946 $\pm$ 0.0319 | 0.0054 $\pm$ 0.0067 |
> | 49.8 | 0.0987 $\pm$ 0.0557 | 0.0051 $\pm$ 0.0064 |
> | 52.3 | 0.0960 $\pm$ 0.0465 | 0.0050 $\pm$ 0.0064 |
>
> The KL divergence stays closely around the target $\epsilon = 0.1$ throughout training, confirming the constraint is  satisfied. The Lagrangian multiplier decreases and stabilizes early on, indicating the dual update converges.
>
> W6 (Off-policy DIME comparison - Q5). We evaluated DIME in its native off-policy setting on HumanoidBench:
>
> | Time | Method | Walk | Run | Hurdle | Balance | Agg |
> | --- | --- | --- | --- | --- | --- | --- |
> | 2h | DIME (off-policy) | 23 $\pm$ 5 | 6 $\pm$ 3 | 6 $\pm$ 2 | 82 $\pm$ 4 | 29 $\pm$ 31 |
> | 2h | **TruDi** | 911 $\pm$ 36 | 905 $\pm$ 9 | 285 $\pm$ 83 | 735 $\pm$ 39 | 709 $\pm$ 255 |
> | 6h | DIME (off-policy) | 862 $\pm$ 60 | 349 $\pm$ 137 | 76 $\pm$ 24 | 203 $\pm$ 80 | 372 $\pm$ 299 |
> | 6h | **TruDi** | **959 $\pm$ 3** | **926 $\pm$ 5** | **836 $\pm$ 47** | **761 $\pm$ 51** | **871 $\pm$ 78** |
> | ~10h | DIME (off-policy) | 852 $\pm$ 191 | 620 $\pm$ 232 | 399 $\pm$ 33 | 375 $\pm$ 142 | 561 $\pm$ 193 |
>
> TruDi at 2h already exceeds DIME (off-policy) at ~10h in aggregated returns.
>
> W7 (Diffusion steps ablation - Q2).
>
> | K | Agg Return | Agg Time (h) |
> | --- | --- | --- |
> | 1 | 15.7 $\pm$ 16.4 | 1.32 $\pm$ 0.44 |
> | 4 | 32.7 $\pm$ 4.0 | 1.58 $\pm$ 0.45 |
> | 8 | **35.3 $\pm$ 3.3** | 1.93 $\pm$ 0.41 |
> | 16 | 35.0 $\pm$ 3.6 | 2.48 $\pm$ 0.45 |
> | 32 | 34.7 $\pm$ 3.8 | 3.71 $\pm$ 0.45 |
>
> Performance saturates at K=8 while training time continues to grow. K=1 fails, confirming the diffusion process is essential. K=8 offers the best performance/cost tradeoff.

---

> > ### Author Rebuttal · Reviewer_auMA · 2026-04-01
> >
> > Thanks to the authors for the detailed rebuttal. I have also read the responses to all other reviewers, which provided helpful additional context. Below is my updated assessment.
> >
> > - W1 (Bound tightness). Having read the response to Reviewer F4A3, I accept that the method only needs a valid upper bound, not a tight one. The epsilon sweep in Fig. 4 supports this: if the bound were vacuously loose, varying epsilon should not produce such a clean and predictable trade-off. I'd still appreciate a brief discussion of this point in the camera-ready.
> >
> > - W2 (Computational overhead). The wall-clock matched comparison (from Reviewer KToi Q1) settles this: TruDi wins at equal time, and REPPO doesn't catch up even with 3x more steps. The K=4 option at 48% overhead is also a practical alternative. I think this has been resolved.
> >
> > - W3 (Standard tasks). The explanation is reasonable, but I maintain that "strong new baseline for massively parallel on-policy RL" oversells the generality.
> >
> > - W4 (Incremental combination). The DPPO/GenPO comparison in Reviewer F4A3's response helped. I'm more positive on this now.
> >
> > - W5–W7. All adequately addressed. The dual variable tracking (W5) and K-ablation (W7) are exactly what I was looking for. The DIME off-policy comparison (W6) is a strong addition.
> >
> > Hence, I am happy to raise my score to 5.

---

> > > ### Author Response · Authors · 2026-04-07
> > >
> > > We thank the reviewer again for the detailed reassessment and for raising the score. Their comments led to several important additions, which have substantially strengthened the paper. We will incorporate this feedback, including a brief discussion on the role of bound tightness, into the camera-ready version.

---

### Official Review · Reviewer_KToi · 2026-03-13

**Soundness:** 3
**Presentation:** 3
**Significance:** 4
**Originality:** 4
**Overall Recommendation:** 5
**Confidence:** 3

**Summary:**

This paper introduces TruDi to address the problem of learning diffusion policies in a massively parallel on-policy RL setting, where Gaussian policies currently dominate. TruDi replaces the intractable marginal KL trust-region constraint with a tractable upper bound defined over the full diffusion trajectory, and uses this to optimize a maximum-entropy actor-critic objective. During evaluation, a probability-flow ODE is used to guide the diffusion policy toward higher-return actions. The method is evaluated extensively across 4 benchmark suites with 73 tasks. The empirical results show that TruDi achieves similar or better performance than the baselines, with the gains becoming more pronounced on the more challenging high-dimensional humanoid settings.

**Compliance With Llm Reviewing Policy:**

Affirmed.

**Final Justification:**

The rebuttal strengthens the paper by resolving my main technical concerns about fairness and attribution of gains. In particular, the matched wall-clock comparison and the trust-region vs. ODE ablation increase my confidence that the reported improvements reflect the core method rather than evaluation or compute differences.

**Key Questions For Authors:**

1. What is the performance of the baselines if they are run for the same wall-clock time budget as TruDi? Is the improvement still present under a matched time budget?
2. You empirically study the multimodality of TruDi on simple symmetric tasks. Do you expect this multimodality to remain useful on more complex problems, such as locomotion with muscles, where many different muscle activation patterns can lead to similar outcomes?
3. Can you isolate the contribution of the trajectory-level KL trust-region surrogate from the contribution of ODE-based evaluation ?

**Limitations:**

yes

**Strengths And Weaknesses:**

Strengths:
- The authors tackle the interesting problem of learning a diffusion policy in the on-policy setting, where trust-region methods are difficult to apply because the marginal likelihood is intractable. The paper proposes a neat way to bypass this by deriving a tractable upper bound on the marginal KL via the full diffusion trajectory, and it achieves strong performance overall.
- The authors extensively evaluate their method on a wide range of tasks, including manipulation and locomotion benchmarks, and compare against both Gaussian and diffusion-based baselines.
- The diffusion policy can represent multimodal action distributions, which may be important for some problems. The paper provides empirical evidence of this on symmetric toy tasks.

Weaknesses:
- It is unclear how much of the performance gain comes from the trust-region upper bound itself and how much comes from the evaluation setup, especially the use of ODE-based evaluation.
- Although the authors discuss wall-clock training time, it remains unclear what would happen if the other methods were given the same time budget rather than the same environment-step budget.

---

> ### Author Rebuttal · Authors · 2026-03-30
>
> We thank Reviewer KToi for the positive and insightful review.
>
> W1 (Trust-region vs. ODE contribution). Please see Q3 below, where we provide a controlled ablation isolating both contributions.
>
> W2 (Wall-clock time fairness). Please see Q1 below, where we compare TruDi and REPPO under matched wall-clock budgets.
>
> Q1 (Matched wall-clock time). We ran REPPO with both matched wall-clock time and extended training (150M steps) on MuJoCo Playground Humanoid tasks:
>
> | Method | G1-Flat | G1-Rough | T1-Flat | T1-Rough | Agg Return | Agg Time (h) |
> | --- | --- | --- | --- | --- | --- | --- |
> | **TruDi (50M steps)** | **38.2 $\pm$ 0.3** | **32.9 $\pm$ 0.5** | **37.8 $\pm$ 0.3** | **31.4 $\pm$ 2.6** | **35.3 $\pm$ 3.3** | 1.93 $\pm$ 0.41 |
> | REPPO (matched time) | 37.4 $\pm$ 0.3 | 31.8 $\pm$ 0.3 | 31.8 $\pm$ 5.1 | 30.2 $\pm$ 1.2 | 32.3 $\pm$ 3.5 | 1.92 $\pm$ 0.44 |
> | REPPO (150M steps) | 37.1 $\pm$ 0.5 | 31.8 $\pm$ 0.2 | 27.2 $\pm$ 7.9 | 29.4 $\pm$ 3.8 | 32.0 $\pm$ 4.7 | 2.78 $\pm$ 0.89 |
>
> TruDi outperforms REPPO even under matched wall-clock time. REPPO at 150M steps (2.78h) performs no better, suggesting that TruDi's advantage comes from the expressiveness of the diffusion policy, not from additional compute.
>
> Q2 (Multimodality on complex tasks). This is an excellent question. We believe multimodality is indeed beneficial on complex tasks, though its value manifests differently than on the symmetric toy tasks. On high-dimensional humanoid control, the benefit is not necessarily about discovering discrete modes (clockwise vs. counter-clockwise), but about maintaining a richer distribution over action sequences during exploration. A Gaussian policy must commit to a single mean, which can get stuck in local optima. The diffusion policy can maintain probability mass across multiple promising regions simultaneously, enabling more robust exploration. Our humanoid results, where TruDi shows the largest gains precisely on the hardest tasks (window cleaning, hurdling, balance), support this hypothesis. Muscle-activation tasks with inherently multi-modal solutions are an exciting direction we plan to explore in future work.
>
> Q3 (Isolating trust-region vs. ODE evaluation). We compared TruDi with and without the trust region (the latter equivalent to DIME on-policy), under both SDE and ODE evaluation on 4 MuJoCo Playground Humanoid tasks:
>
> | Method | Eval | G1-Flat | G1-Rough | T1-Flat | T1-Rough | Avg |
> | --- | --- | --- | --- | --- | --- | --- |
> | TruDi (w/o TR) | SDE | 10.4 $\pm$ 6.8 | 9.6 $\pm$ 3.0 | 27.5 $\pm$ 3.6 | 13.1 $\pm$ 7.2 | 13.5 $\pm$ 4.6 |
> | **TruDi** | SDE | 31.6 $\pm$ 0.2 | 26.4 $\pm$ 0.2 | 34.1 $\pm$ 0.9 | 24.9 $\pm$ 3.5 | 29.6 $\pm$ 2.4 |
> | TruDi (w/o TR) | ODE | 17.0 $\pm$ 5.3 | 17.2 $\pm$ 2.9 | 33.7 $\pm$ 2.6 | 18.8 $\pm$ 8.6 | 19.8 $\pm$ 4.5 |
> | **TruDi** | ODE | **38.2 $\pm$ 0.3** | **32.9 $\pm$ 0.5** | **37.8 $\pm$ 0.3** | **31.4 $\pm$ 2.6** | **35.5 $\pm$ 2.1** |
> - TR means trust region
>
> Our ablation separates the two contributions clearly. The trust region is the dominant factor: adding it improves the average return from 13.5 to 29.6 under SDE evaluation, and from 19.8 to 35.5 under ODE evaluation. ODE evaluation also contributes meaningfully on its own, improving TruDi from 29.6 to 35.5 and TruDi w/o trust region (DIME on-policy) from 13.5 to 19.8. Both contributions are independent and complementary.

---

> > ### Author Rebuttal · Reviewer_KToi · 2026-04-03
> >
> > The rebuttal resolves my main concerns. The matched wall-clock results address the fairness issue, and the new ablation clarifies that the trust region is the main contributor, with ODE evaluation providing an additional gain. The multimodality discussion would still benefit from more validation on complex tasks, but this is not a major remaining concern.

---

> > > ### Author Response · Authors · 2026-04-07
> > >
> > > We are glad that the new experiments have resolved the reviewer’s main concerns, and these analyses will be included in the final version. We are also planning to further evaluate TruDi on the task suggested by the reviewer, namely the muscle-control setting, to better test its multimodal behavior.

---

### Official Review · Reviewer_F4A3 · 2026-03-16

**Soundness:** 3
**Presentation:** 3
**Significance:** 3
**Originality:** 3
**Overall Recommendation:** 4
**Confidence:** 3

**Summary:**

This submission introduces a trust-region based method for training diffusion policies in massively parallel, on-policy reinforcement learning. The main technical contribution is a tractable trust-region formulation: because the marginal action likelihood of a diffusion policy is intractable, the authors instead constrain the KL divergence over the full diffusion trajectory, yielding a tractable surrogate for stable on-policy updates. Built on a maximum-entropy actor-critic framework, the proposed method also uses a probability-flow ODE for deterministic evaluation, and is evaluated on 4 benchmark suites with 73 tasks, where it is reported to match strong Gaussian on-policy baselines on standard control tasks and outperform them on more challenging high-dimensional humanoid settings.

**Compliance With Llm Reviewing Policy:**

Affirmed.

**Final Justification:**

My major concerns have been addressed by the authors' rebuttal.

**Key Questions For Authors:**

1. Can you further clarify novelty relative to prior on-policy diffusion RL, especially DPPO and GenPO?
2. Can you clarify fairness and reproducibility details in the comparisons? In particular: are the compared methods capacity-matched, given the different actor/critic architectures and critic types; and can you resolve the inconsistencies in implementation details such as TD(𝜆) vs. “GAE 𝜆” and the differing critic activation descriptions?

**Limitations:**

Please refer to the Strengths and Weaknesses section for details.

A minor one:

There seems to be inconsistency in the paper’s positioning. Early in the introduction it says training diffusion policies from scratch with on-policy RL “has not been researched in the literature,” but later the related-work section explicitly lists DPPO and GenPO, with GenPO described as enabling learning from scratch.

**Strengths And Weaknesses:**

Strengths

1.  Interesting technical idea. The paper proposes a tractable trust-region constraint over the full diffusion trajectory, which is a principled way to handle the intractable marginal likelihood issue in diffusion policies.
2. Extensive empirical efforts and good results. Reported experiments cover 4 benchmark suites and a variety of tasks, with IQM and confidence interval over 10 seeds. The results of the proposed method are competitive with sota on standard tasks and outperform sota on challenging high-dimensional humanoid control.

Weaknesses

1. While the core formula is motivated from the intractability of marginal likelihood in diffusion policy, it still lacks information or analysis about how tight and practically meaningful this upper bound is.
2. The ablation study has not fully isolated the central technical new idea, i.e., full-trajectory trust-region surrogate. In particular, the implementation contains several other design choices, including a REPPO-style actor-critic recipe, dual trust-region optimization, and specific architecture/critic design choices (say several baselines are not capacity-matched in Table 4), making it hard to tell how much of the performance gain comes specifically from the full-trajectory trust-region formulation versus other optimization/design decisions.

---

> ### Author Rebuttal · Authors · 2026-03-30
>
> We thank Reviewer F4A3 for the positive assessment and thoughtful questions.
>
> W1 (Bound tightness). We thank the reviewer for raising this important point. We would like to highlight that we do not claim that the trajectory-level trust-region bound is generally tight. As shown in Appendix A (Eq. 28--29), the gap between the marginal trust-region KL and our trajectory-level KL is given by the conditional KL over denoising trajectories conditioned on the same output action $a^0$; for stochastic diffusion policies, this term is generally nonzero, so tightness is difficult to characterize in full generality. We also recognize that Eq. 29 contains a sign typo, and the decomposition should involve a minus sign rather than a plus sign in the second term; we will correct this in the camera-ready version. Importantly, our method only requires a valid upper bound on the marginal KL, which is exactly what Eq. 28 provides, while our main focus is the empirical role of the trust-region constraint in stabilizing training. This is reflected in Fig. 4 (Left): overly strict bounds ($\epsilon = 0.01$) lead to slow updates, while overly loose bounds ($\epsilon \geq 10$) remove useful regularization; moderate values $\epsilon \in [0.1, 0.4]$ yield the best performance across a broad range, indicating that the trajectory-level trust region is effective in practice. We will add this discussion and correct the typo in Eq. 29 in the camera-ready version.
>
> W2 (Isolating the trust-region contribution). We acknowledge that TruDi builds on REPPO's actor-critic recipe. This is deliberate: by sharing the same optimization framework, we isolate the effect of the policy parameterization (diffusion vs. Gaussian). The SDE vs. ODE ablation in our response to Reviewer KToi (Q3) further decouples the evaluation strategy from the training algorithm.
>
> Q1 (Novelty relative to DPPO and GenPO). Unlike DPPO and GenPO, which primarily make diffusion compatible with PPO-style clipped updates, TruDi introduces a trajectory-level trust-region MaxEnt formulation. Its key novelty is the tractable upper bound on the marginal KL over the full diffusion trajectory, together with a Q-based MaxEnt actor-critic update for massively parallel on-policy RL. Specifically:
>
> - DPPO employs a multi-layer MDP view, applying per-step PPO clipping to each denoising step independently. TruDi derives a principled upper bound on the marginal KL via the full trajectory, with automatic dual variable tuning.
> - GenPO constructs an invertible flow mapping via a doubled dummy action space to compute exact likelihoods, requiring doubled action dimensionality and a compression loss to mitigate spurious exploration. GenPO also employs a PPO-like clipping objective for handling the trust region. TruDi avoids this overhead entirely by working directly with the tractable trajectory KL bound, requiring no action space modification.
> - Neither DPPO nor GenPO incorporates a learned Q-function for pathwise policy gradients; both rely on GAE-based advantage estimation with score-function gradients.
>
> Q2 (Fairness and reproducibility). We will clarify these in the revision:
>
> - TD($\lambda$) vs. GAE: GAE is used by PPO-based methods to estimate advantages from a state-value function, while TD($\lambda$) is used by REPPO-based methods [3] (including TruDi) to compute targets for the learned Q-function.
> - Critic activations: GeLU is used for TruDi and REPPO [3]; ELU for PPO following [2].
> - Capacity: Regarding capacity matching (Table 4 in the paper): the diffusion policy uses larger networks (512 vs. 256). To investigate whether simply increasing network size explains TruDi's gains, we ran PPO with different hidden sizes on the DMC benchmark:
>
> | Method | Agg IQM |
> | --- | --- |
> | PPO h=256 | **671.6** $\pm$ **246.1** |
> | PPO h=512 | 618.7 $\pm$ 247.6 |
> | PPO h=1024 | 534.2 $\pm$ 270.7 |
> - h means hidden latent size
>
> Larger networks do not improve PPO performance; in fact, they hurt it. This is consistent with findings in [1]. Regarding activation functions, we keep the default activations from [2] for each method to ensure reproducibility.
>
> Positioning inconsistency. We thank the reviewer for catching this. Indeed, the related works section correctly discusses DPPO and GenPO as concurrent efforts in this direction. We will revise the introduction to align with the related works section and more precisely articulate TruDi's distinct contribution: the principled trust-region formulation over the full diffusion trajectory within the MaxEnt RL framework.
>
> [1] M. Andrychowicz et al., "What Matters In On-Policy Reinforcement Learning? A Large-Scale Empirical Study,"
>
> [2] Schwarke, C. et al. RSL-RL: A learning library for robotics research.
>
> [3] Voelcker, C. et al. Relative entropy pathwise policy optimization.

---

> > ### Author Rebuttal · Reviewer_F4A3 · 2026-04-06
> >
> > I would like to thank the authors for addressing my main concerns. It would be helpful to incorporate them into the final veresion.

---

> > > ### Author Response · Authors · 2026-04-07
> > >
> > > We would like to thank again the reviewer for the thoughtful and constructive feedback, which has helped us sharpen several important parts of the paper. The final version will incorporate these discussions as well as the additional experiments.

---

### Decision · Program_Chairs · 2026-04-30

**Decision:**

Accept (regular)

**Comment:**

The paper introduces trust-region diffusion policies (TruDi), a novel framework that bridges the gap between the expressive multimodal representation of diffusion models and the stability requirements of on-policy, massively parallel reinforcement learning by deriving a tractable upper bound for the KL-divergence over the entire diffusion trajectory. Reviewers generally lauded the technical elegance of the trajectory-level trust region and the comprehensive empirical evaluation across 73 tasks, though initial concerns were raised regarding the tightness of the KL bound and the computational overhead relative to Gaussian baselines. The authors’ rebuttal successfully addressed these points by providing new ablations—demonstrating that TruDi outperforms state-of-the-art methods like REPPO even under matched wall-clock time and that the trust-region constraint is the primary driver of stability and performance in high-dimensional humanoid tasks. Given the solid theoretical foundation, the resolution of fairness concerns, and the significant performance gains in complex control settings, I recommend the paper for acceptance.